# Bandits with Stochastic Corruption:
# Lower Bounds on Regret and Robust Optimistic Algorithms

**Timothée Mathieu**                                    *timothee.mathieu@inria.fr*

**Debabrota Basu**                                      *debabrota.basu@inria.fr*

**Odalric-Ambrym Maillard**                             *odalric.maillard@inria.fr*

*Université de Lille, Inria, CNRS, Centrale Lille UMR 9189 – CRIStAL, F-59000 Lille, France*

**Reviewed on OpenReview:** *https://openreview.net/forum?id=oGIROic3jU*

## Abstract

We study the Bandits with Stochastic Corruption problem, i.e. a stochastic multi-armed bandit problem with $k$ unknown reward distributions, which are heavy-tailed and corrupted by a history-independent stochastic adversary or Nature. To be specific, the reward obtained by playing an arm comes from corresponding heavy-tailed reward distribution with probability $1 - \varepsilon \in (0.5, 1]$ and an arbitrary corruption distribution of unbounded support with probability $\varepsilon \in [0, 0.5)$. First, we provide *a problem-dependent lower bound on the regret* of any corrupted bandit algorithm. The lower bounds indicate that the Bandits with Stochastic Corruption problem is harder than the classical stochastic bandit problem with sub-Gaussian or heavy-tail rewards. Following that, we propose a novel UCB-type algorithm for Bandits with Stochastic Corruption, namely `HuberUCB`, that builds on Huber's estimator for robust mean estimation. Leveraging a novel concentration inequality of Huber's estimator, we prove that `HuberUCB` achieves a near-optimal regret upper bound. Since computing Huber's estimator has quadratic complexity, we further introduce a sequential version of Huber's estimator that exhibits linear complexity. We leverage this sequential estimator to design `SeqHuberUCB` that enjoys similar regret guarantees while reducing the computational burden. Finally, we experimentally illustrate the efficiency of `HuberUCB` and `SeqHuberUCB` in solving Bandits with Stochastic Corruption for different reward distributions and different levels of corruptions.

## 1 Introduction

The multi-armed bandit problem is an archetypal setting to study sequential decision-making under incomplete information (Lattimore & Szepesvári, 2020). In the classical setting of stochastic multi-armed bandits, the decision maker or agent has access to $k \in \mathbb{N}$ unknown reward distributions or arms. At every step, the agent plays an arm and obtains a reward. The goal of the agent is to maximize the expected total reward accumulated by a given horizon $T \in \mathbb{N}$.

In this paper, we are interested in a challenging extension of the classical multi-armed bandit problem, where the reward at each step may be corrupted by Nature, which is a stationary mechanism independent of the agent's decisions and observations. This setting is often referred as the *Corrupted Bandits*. Specifically, we extend the existing studies of corrupted bandits (Lykouris et al., 2018; Bogunovic et al., 2020; Kapoor et al., 2019) to the more general case, where the 'true' reward distribution might be heavy-tailed (i.e. with a finite number of finite moments) and the corruption can be unbounded.

**Bandits with Stochastic Corruption.** Specifically, we model a corrupted reward distribution as $(1-\varepsilon)P + \varepsilon H$, where $P$ is the distribution of inliers with a finite variance, $H$ is the distributions of outliers with possibly

unbounded support, and $\varepsilon \in [0, 1/2)$ is the proportion of outliers. Thus, in the corresponding stochastic bandit setting, an agent has access to $k$ arms of corrupted reward distributions $\{(1-\varepsilon)P_i + \varepsilon H_i\}_{i=1}^k$. Here, $P_i$'s are uncorrupted reward distributions with heavy-tails and bounded variances, and $H_i$'s are corruption distributions with possibly unbounded corruptions. The goal of the agent is to maximize the expected total reward accumulated oblivious to the corruptions. This is equivalent to considering a setting where at every step Nature flips a coin with success probability $\varepsilon$. The agent obtains a corrupted reward if Nature obtains 1 and otherwise, an uncorrupted reward. We call this setting *Bandits with Stochastic Corruption* as the corruption introduced in each step does not depend on the present or previous choices of arms and observed rewards. Our setting encompasses both heavy-tailed rewards and unbounded corruptions. We formally define the setting and corresponding regret definition in Section 3.

Though this article primarily focuses on the theoretical understanding of the interplay between corruption, heavytailedness and decision making, we find it relevant to pinpoint at a few applications for which this setting may apply. Note that heavy-tail distributions are naturally motivated by applications in economy and financial markets (Agrawal et al., 2021), while corrupted distributions are naturally motivated by robustness issues in life sciences and applications in games or security, or when dealing with a misspecified model (Hotho, 2022; Golrezaei et al., 2021; Singh & Upadhyaya, 2012). Hence the combination of corrupted and heavy-tail distributions naturally appears at the crossing of these classical application domains.

Bandits with Stochastic Corruption is different from the adversarial bandit setting (Auer et al., 2002). The adversarial bandit assumes existence of a non-stochastic adversary that can return at each step the worst-case reward to the agent depending on its history of choices. Incorporating corruptions in this setting, Lykouris et al. (2018) and Bogunovic et al. (2020) consider settings where the rewards can be corrupted by a history-dependent adversary but the total amount of corruption and also the corruptions at each step are bounded. In contrast to the adversarial corruption setting in the literature, we consider a non-adversarial proportion of corruptions ($\varepsilon \in [0, 1/2)$) at each step, which are stochastically generated from unbounded corruption distributions $\left(\{H_i\}_{i=1}^k\right)$. To the best of our knowledge, only Kapoor et al. (2019) have studied similar non-adversarial corruption setting with a history-independent proportion of corruption at each step for regret minimization. But they assume that the probable corruptions at each step are bounded, and the uncorrupted rewards are sub-Gaussian. On the other hand, Altschuler et al. (2019) study the same stochastic unbounded corruption that we consider but they focus on best arm identification using the median of the arms as a goal making this a different problem. Hence, we observe that *there is a gap in the literature in studying unbounded stochastic corruption for bandits with possibly heavy-tailed rewards and this article aims to fill this gap.* Specifically, we aim to deal with unbounded corruption and heavy-tails simultaneously, which requires us to develop a novel sensitivity analysis of the robust estimator in lieu of a worst-case (adversarial bandits) analysis.

**Our Contributions.** Specifically, in this paper, we aim to investigate three main questions:

1. Is the setting of Bandits with Stochastic Corruption with unbounded corruptions and heavy tails fundamentally harder (in terms of the regret lower bound) than the classical sub-Gaussian and uncorrupted bandit setting?
2. Is it possible to design an *efficient and robust algorithm* that achieves an order-optimal performance (*logarithmic* regret) in the stochastic corruption setting?
3. Are robust bandit algorithms *efficient in practice*?

These questions have led us to the following contributions:

*1. Hardness of Bandits with Stochastic Corruption with unbounded corruptions and heavy tails.* In order to understand the fundamental hardness of the proposed setting, we use a suitable notion of regret (Kapoor et al., 2019), denoted by $\mathfrak{R}_n$ (Definition 1), that extends the traditional pseudo-regret (Lattimore & Szepesvári, 2020) to the corrupted setting. Then, in Section 4, we derive lower bounds on regret that reveal increased difficulties of corrupted bandits with heavy tails in comparison with the classical non-corrupted and light-tailed Bandits. (a) In the heavy-tailed regime (3), we show that even when the suboptimality gap

$\Delta_i$[1] for arm $i$ is large, the regret increases with $\Delta_i$ because of the difficulty to distinguish between two arms when the rewards are heavy-tailed. (b) Our lower bounds indicate that when $\Delta_i$ is large, the logarithmic regret is asymptotically achievable, but the hardness depends on the corruption proportion $\varepsilon$, variance of $P_i$, denoted by $\sigma_i^2$, and the suboptimality gap $\Delta_i$. Specifically, if $\frac{\Delta_i}{\sigma_i}$'s are small, i.e. we are in low distinguishability/high variance regime, the hardness is dictated by $\frac{\sigma_i^2}{\overline{\Delta}_{i,\varepsilon}^2}$. Here, $\overline{\Delta}_{i,\varepsilon} \triangleq \Delta_i(1-\varepsilon) - 2\varepsilon\sigma_i$ is the '*corrupted suboptimality gap*' that replaces the traditional suboptimality gap $\Delta_i$ in the lower bound of non-corrupted and light-tailed bandits (Lai & Robbins, 1985). Since $\overline{\Delta}_{i,\varepsilon} \leq \Delta_i$, it is harder to distinguish the optimal and suboptimal arms in the corrupted settings. They are the same when the corruption proportion $\varepsilon = 0$. In this article, we exclude the case when $\overline{\Delta}_{i,\varepsilon} \leq 0$ as it essentially corresponds to the case when corruption is large enough to render reward distributions hard to distinguish. Hence we limit our study to the case when $\overline{\Delta}_{i,\varepsilon} > 0$.

Additionally, our analysis partially addresses an open problem in heavy-tailed bandits. Works on heavy-tailed bandits (Bubeck et al., 2013; Agrawal et al., 2021) rely on the assumption that a bound on the $(1+\eta)$-moment, i.e. $\mathbb{E}[|X|^{1+\eta}]$, is known for some $\eta > 0$. We do not assume such a restrictive bound, as knowing a bound on $\mathbb{E}[|X|^{1+\eta}]$ implies the knowledge of a bound on the differences between the means of the reward of the different arms. Instead, we assume that the centered moment, specifically the variance, is bounded by a known constant. Thus, we partly address the open problem of Agrawal et al. (2021) by relaxing the classical bounded $(1+\eta)$-moment assumption with the bounded centered moment one, for $\eta \geq 1$.

*2. Robust and Efficient Algorithm Design.* In Section 5, we propose a robust algorithm, called `HuberUCB`, that leverages the Huber's estimator for robust mean estimation using the knowledge of $\varepsilon$ and a bound on the variances of inliers. We derive a novel concentration inequality on the deviation of empirical Huber's estimate that allows us to design robust and tight confidence intervals for `HuberUCB`. In Theorem 3, we show that `HuberUCB` achieves the logarithmic regret, and also the optimal rate when the sub-optimality gap $\Delta$ is not too large. We show that for `HuberUCB`, $\mathfrak{R}_n$ can be decomposed according to the respective values of $\Delta_i$ and $\sigma_i$:

$$
\mathfrak{R}_n \leq \underbrace{\mathcal{O}\left(\sum_{i:\Delta_i > \sigma_i} \log(n)\sigma_i\right)}_{\text{Error due to heavy-tail}} + \underbrace{\mathcal{O}\left(\sum_{i:\Delta_i \leq \sigma_i} \log(n)\Delta_i \frac{\sigma_i^2}{\overline{\Delta}_{i,\varepsilon}^2}\right)}_{\sigma^2/\Delta \text{ error with corrupted sub-optimality gaps}}.
$$

Thus, our upper bound allows us to segregate the errors due to heavy-tail, corruption, and corruption-correction with heavy tails. The error incurred by `HuberUCB` can be directly compared to the lower bounds obtained in Section 4 and interpreted in both the high distinguishibility regime and the low distinguishibility regime as previously mentioned.

*3. Empirically Efficient and Robust Performance.* To the best of our knowledge, we present the first robust mean estimator that can be computed in a linear time in a sequential setting (Section 6). Existing robust mean estimators, such as Huber's estimator, need to be recomputed at each iteration using all the data, which implies a quadratic complexity. Our proposal recomputes Huber's estimator only when the iteration number is a power of 2 and computes a sequential approximation on the other iterations. We use the Sequential Huber's estimator to propose `SeqHuberUCB`. We theoretically show that `SeqHuberUCB` achieves similar order of regret as `HuberUCB`, while being computationally efficient. In Section 7, we also experimentally illustrate that `HuberUCB` and `SeqHuberUCB` achieve the claimed performances for corrupted Gaussian and Pareto environments.

We further elaborate on the novelty of our results and position them in the existing literature in Section 2. For brevity, we defer the detailed proofs and the parameter tuning to Appendix.

---

[1]The suboptimality gap of an arm is the difference in mean rewards of an optimal arm and that arm. In our context suboptimality gap refer to the gap between the inlier distributions which can be compared to suboptimality gap in heavy-tail setting, as opposed to the corrupted gap that we define later.

## 2 Related Work

Due to the generality of our setting, this work either extends or relates to the existing approaches in both the heavy-tailed and corrupted bandits literature. While designing the algorithm, we further leverage the literature of robust mean estimation. In this section, we connect to these three streams of literature. Table 1 summarizes the previous works and posits our work in lieu.

| Algorithms | Settings | Corruption | Type of outliers | Heavy-tailed | Adversarial/ Stochastic |
|---|---|---|---|---|---|
| Our work | MAB | Yes | Unbounded | Yes | Stochastic |
| Bubeck et al. (2013); Agrawal et al. (2021); Lee et al. (2020) | MAB | No | x | Yes | Stochastic |
| Lykouris et al. (2018) | MAB | Yes | Bounded | No | Stochastic |
| Bogunovic et al. (2020) | GP Bandits | Yes | Bounded | No | Adversarial |
| Kapoor et al. (2019) | MAB & Linear Bandits | Yes | Bounded | No | Stochastic |
| Medina & Yang (2016); Shao et al. (2018) | Linear Bandits | No | x | Yes | Stochastic |
| Bouneffouf (2021) | Contextual Bandits | context only | Unbounded | No | Stochastic |
| Agarwal et al. (2019) | Control | Yes | Bounded | x | Adversarial |
| Hajiesmaili et al. (2020); Auer et al. (2002); Pogodin & Lattimore (2020) | MAB | Yes | Bounded | x | Adversarial |

Table 1: Comparison of existing results on Corrupted and Heavy-tailed Bandits.

*Heavy-tailed bandits.* Bubeck et al. (2013) are one of the first to study robustness in multi-armed bandits by studying the heavy-tailed rewards. They use robust mean estimator to propose the RobustUCB algorithms. They show that under assumptions on the raw moments of the reward distributions, a logarithmic regret is achievable. It sprouted research works leading to either tighter rates of convergence (Lee et al., 2020; Agrawal et al., 2021), or algorithms for structured environments (Medina & Yang, 2016; Shao et al., 2018). Our article uses Huber's estimator which was already discussed in Bubeck et al. (2013). However, the chosen parameters in Bubeck et al. (2013) were suited for heavy-tailed distributions, and thus, *render their proposed estimator non-robust to corruption. We address this gap in this work.*

*Corrupted bandits.* The existing works on Corrupted Bandits (Lykouris et al., 2018; Bogunovic et al., 2020; Kapoor et al., 2019) are restricted to bounded corruption. When dealing with bounded corruption, one can use techniques similar to adversarial bandits (Auer et al., 2002) to deal with an adversary that can't corrupt an arm too much. The algorithms and proof techniques are fundamentally different in our article because the stochastic (or non-adversarial) corruption by Nature allows us to learn about the inlier distribution on the condition that corresponding estimators are robust. Thus, *our bounds retain the problem-dependent regret, while successfully handling probably unbounded corruptions with robust estimators.*

*Robust mean estimation.* Our algorithm design leverages the rich literature of robust mean estimation, specifically the influence function representation of Huber's estimator. The problem of robust mean estimation in a corrupted and heavy-tailed setting stems from the work of Huber (Huber, 1964; 2004). Recently, in tandem with machine learning, there have been numerous advances both in the heavy-tailed (Devroye et al., 2016; Catoni, 2012; Minsker, 2019), and in the corrupted settings (Lecué & Lerasle, 2020; Minsker & Ndaoud, 2021; Prasad et al., 2019; 2020; Depersin & Lecué, 2022; Lerasle et al., 2019; Lecué & Lerasle, 2020). Our work, specifically the novel concentration inequality for Huber's estimator, enriches this line of work with a result of parallel interest. We also introduce a sequential version of Huber's estimator achieving linear complexity.

## 3   Bandits with Stochastic Corruption: Problem formulation

In this section, we present the corrupted bandits setting that we study, together with the corresponding notion of regret. Similarly to the classical bandit setup, the regret decomposition lemma allows us to focus on the expected number of pulls of a suboptimal arm as the central quantity to control algorithmic standpoint.

**Notations.** We denote by $\mathcal{P}$ the set of probability distributions on the real line $\mathbb{R}$ and by $\mathcal{P}_{[q]}(M) \triangleq \{P \in \mathcal{P} : \mathbb{E}_P[|X - \mathbb{E}_P[X]|^q] \leq M\}$ the set of distributions with $q^{th}$ moment, $q \geq 1$, bounded by $M > 0$. $\mathbf{1}\{A\}$ is the indicator function for the event $A$ being true. We denote the mean of a distribution $P_i$ as $\mu_i \triangleq \mathbb{E}_{P_i}[X]$. For any $\mathcal{D} \subset \mathcal{P}$, we denote $\mathcal{D}(\varepsilon) \triangleq \{(1-\varepsilon)P + \varepsilon H : P \in \mathcal{D}, H \in \mathcal{P}\}$ the set of corrupted distributions from $\mathcal{D}$.

**Problem Formulation.**   In the setting of *Bandits with Stochastic Corruption*, a bandit algorithm faces an environment with $k \in \mathbb{N}$ many reward distributions in the form $\nu^\varepsilon = (\nu_i^\varepsilon)_{i=1}^k$ where $\nu_i^\varepsilon = (1 - \varepsilon)P_i + \varepsilon H_i$ denotes the distribution of rewards of arm $i$. Here $P_i, H_i$ are real-valued distributions and $\varepsilon$ is a mixture parameter assumed to be in $[0, 1/2)$, that is $P_i$ is given more weights than $H_i$ in the mixture of arm $i$. For this reason, the $\{P_i\}_{i=1}^k$ are called the *inlier* distributions and the $\{H_i\}_{i=1}^k$ the *outlier* distributions. We assume the inlier distributions have at least 2 finite moments that is $P_1, \ldots, P_k \in \mathcal{P}_{[2]}(M)$ for some $M > 0$, while no restriction is put on the outlier distributions, that is $H_1, \ldots, H_k \in \mathcal{P}$. For this reason, we also refer to the outlier distributions as the *corrupted* distributions, and to the inlier distributions as the *non-corrupted* ones. $\varepsilon$ is called the level of corruption. For convenience, we further denote by $\nu$ in lieu of $\nu^0 = (P_i)_{i=1}^k$ the reward distributions of the non-corrupted environment.

The game proceed as follows: At each step $t \in \{0, \ldots, n\}$, the agent policy $\pi$ interacts with the corrupted environment by choosing an arm $A_t$ and obtaining a stochastically corrupted reward. To generate this reward, Nature first draws a random variable $C_t \in \{0, 1\}$ from a Bernoulli distribution with mean $\varepsilon \in [0, 1/2)$. If $C_t = 1$, it generates a corrupted reward $Z_t$ from distribution $H_{A_t}$ corresponding to the chosen arm $A_t \in \{1, \ldots, k\}$. Otherwise, it generates a non-corrupted $X_t'$ from distribution $P_{A_t}$. More formally, Nature generates reward $X_t = X_t' \mathbf{1}\{C_t = 0\} + Z_t \mathbf{1}\{C_t = 1\}$ which the learner observes. The learner leverages this observation to choose another arm at the next step in order to maximize the total cumulative reward obtained after $n$ steps. In Algorithm 1, we outline a pseudocode of this framework.

---

**Algorithm 1** Bandits with Stochastic Corruption

---

**Require:** $\varepsilon \in [0, 1/2)$, $q \geq 2$ and $M > 0$
 1: **Input:** $P_1, \ldots, P_k \in \mathcal{P}_{[q]}(M)$ be the uncorrupted reward distributions and $H_1, \ldots, H_k \in \mathcal{P}$ be the corrupted reward distributions.
 2: **for** $t = 1, \ldots, n$ **do**
 3:     Player plays an arm $A_t \in \{1, \ldots, k\}$
 4:     Nature draws a Bernoulli $C_t \sim \text{Ber}(\varepsilon)$
 5:     Generate a corrupted reward $Z_t \sim H_{A_t}$ and an uncorrupted reward $X_t' \sim P_{A_t}$
 6:     Player observe the reward $X_t = X_t' \mathbf{1}\{C_t = 0\} + Z_t \mathbf{1}\{C_t = 1\}$
 7: **end for**

---

**Remark 1 (Non-adversarial corruption.)** *In the setting of Bandits with Stochastic Corruption, we consider that the reward received by the learner is corrupted when $C_t = 1$ and non-corrupted otherwise. Since the law of $C_t$ is a Bernoulli $Ber(\varepsilon)$, the corruption is stochastic, and independent on other variables. This is in contrast with* adversarial *setups, where corruption is typically chosen by an opponent and possibly depending on other variables. Assuming a non-adversarial behavior of the Nature seems more justified than assuming an adversarial setup in applications, such as agriculture where corruption is often due to external disturbances, such as pests appearance or weather hazards, whose occurrence are typically non-adversarial. Now when corruption happens, we do not put restriction on the level of corruption. For example, we can imagine a pest outburst or hail, that may have huge impact on a crop but does not occur adversarially.*

**Remark 2 (Weak assumption on inliers)** *Let us highlight that we do not assume sub-Gaussian behavior for the inlier distributions $P_i$. Instead, we consider only a weak moment assumption, i.e. the inlier distri-*

butions $P_i$ have a finite variance. *Thus, our setting is capable of modeling both the moderately heavy-tailed setting and the corrupted settings. We highlight this generality in the regret lower bounds and empirical performance analysis in Section 4 and 7.*

**Corrupted regret.** In this setting, we observe that a corrupted reward distribution $((1-\varepsilon)P_i+\varepsilon H_i)$ might not have a finite mean, unlike the true $P_i$'s. Thus, the classical notion of regret with respect to the corrupted reward distributions might fail to quantify the goodness of the policy and its immunity to corruption while learning. On the other hand, in this setup, the natural notion of expected regret is measured with respect to the mean of the non-corrupted environment $\nu$ specified by $\{P_i\}_{i=1}^k$.

**Definition 1 (Corrupted Regret)** *In Bandits with Stochastic Corruption, we define the regret of a learning algorithm playing strategy $\pi$ after $n$ steps of interaction with the environment $\nu^\varepsilon$ against constantly playing an optimal arm $\star \in \arg\min_i \mathbb{E}_{P_i}[X']$ as*

$$\mathfrak{R}_n(\pi, \nu^\varepsilon) \triangleq n \max_i \mathbb{E}_{P_i}[X'] - \mathbb{E}\left[\sum_{t=1}^n X'_t\right]. \qquad \text{(Corrupted regret)}$$

*We call this quantity the pseudo-regret under corrupted observation, or for short, the **corrupted regret**.*

The expectation is crucially taken on $X'_i \sim P_i$ and $X'_t \sim P_{A_t}$ but not on $X_i$ and $X_t$. The expectation on the right also incorporates possible randomization from the learner. Thus, (Corrupted regret) quantifies the loss in the rewards accumulated by policy $\pi$ from the inliers while learning only from the *corrupted rewards* and also not knowing the arm with the best *true reward* distribution. Thus, this definition of corrupted regret quantifies the rate of learning of a bandit algorithm as regret does for non-corrupted bandits. A similar notion of regret is considered in (Kapoor et al., 2019) that deals with bounded stochastic corruptions.

Due to the non-adversarial nature of the corruption, the regret can be decomposed, as in classical stochastic bandits, to make appear the expected number of pulls of suboptimal arms $\mathbb{E}_{\nu^\varepsilon}[T_i(n)]$, which allow us to focus the regret analysis on bounding these terms.

**Lemma 1 (Decomposition of corrupted regret)** *In a corrupted environment $\nu^\varepsilon$, the regret writes*

$$\mathfrak{R}_n(\pi, \nu^\varepsilon) = \sum_{i=1}^k \Delta_i \mathbb{E}_{\nu^\varepsilon}[T_i(n)],$$

*where $T_i(n) \triangleq \sum_{t=1}^n \mathbf{1}\{A_t = i\}$ denotes the number of pulls of arm $i$ until time $n$ and the problem-dependent quantity $\Delta_i \triangleq \max_j \mu_j - \mu_i$ is called the suboptimality gap of arm $i$.*

## 4 Lower bounds for uniformly good policies under heavy-tails and corruptions

In order to derive the lower bounds, it is classical to consider *uniformly good* policies on some family of environments, Lai & Robbins (1985). We introduce below the corresponding notion for corrupted environments with the set of laws $\mathfrak{D}^{\otimes k} = \mathcal{D}_1 \otimes \cdots \otimes \mathcal{D}_k$, where $\mathcal{D}_i \subset \mathcal{P}$ for each $i \in \{1, \ldots, k\}$.

**Definition 2 (Robust uniformly good policies)** *Let $\mathfrak{D}^{\otimes k}(\varepsilon) = \mathcal{D}_1(\varepsilon) \otimes \cdots \otimes \mathcal{D}_k(\varepsilon)$ be a family of corrupted bandit environments on $\mathbb{R}$. For a corrupted environment $\nu^\varepsilon \in \mathfrak{D}^{\otimes k}(\varepsilon)$ with corresponding uncorrupted environment $\nu$, let $\mu_i(\nu)$ denote the mean reward of arm $i$ in the uncorrupted setting and $\mu_\star(\nu) \triangleq \max_a \mu_i(\nu)$ denote the maximum mean reward. A policy $\pi$ is uniformly good on $\mathfrak{D}^{\otimes k}(\varepsilon)$ if for any $\alpha \in (0, 1]$,*

$$\forall \nu \in \mathfrak{D}^{\otimes k}(\varepsilon), \forall i \in \{1, \ldots, k\}, \mu_i(\nu) < \mu_\star(\nu) \Rightarrow \quad \mathbb{E}_{\nu^\varepsilon}[T_i(n)] = o(n^\alpha).$$

Since the corrupted setup is a special case of stochastic bandits, a lower bound can be immediately recovered with classical results, such as Lemma 2 below, that is a version of the change of measure argument (Burnetas & Katehakis, 1997), and can be found in (Maillard, 2019, Lemma 3.4).

**Lemma 2 (Lower bound for uniformly good policies)** *Let $\mathfrak{D}^{\otimes k} = \mathcal{D}_1 \otimes \cdots \otimes \mathcal{D}_k$, where $\mathcal{D}_i \subset \mathcal{P}$ for each $i \in \{1, \dots, k\}$ and let $\nu \in \mathfrak{D}^{\otimes k}$. Then, any uniformly good policy on $\mathfrak{D}^{\otimes k}$ must pull arms such that,*

$$\forall i \in \{1, \dots, k\}, \mu(\nu_i) \leq \mu_\star(\nu) \quad \Rightarrow \quad \liminf_{n \to \infty} \frac{\mathbb{E}_\nu[T_i(n)]}{\log(n)} \geq \frac{1}{\mathcal{K}_i(\nu_i, \mu(P_\star))}.$$

*with $\mathcal{K}_i(\nu_i, \mu(P_\star)) = \inf\{D_{\mathrm{KL}}(\nu_i, P^*) : \nu_i \in \mathcal{D}_i, \mu(\nu_i) \geq \mu(P_\star)\}$ where $D_{\mathrm{KL}}$ denotes the Kullback-Leibler divergence between distributions.*

Lemma 2 is used in the traditional bandit literature to obtain lower bound on the regret using the decomposition of regret from Lemma 1. In our setting, however, the lower bound is more complex as it involves optimization on $\mathcal{P}_{[2]}(M)$, the set of distributions with a variance bounded by $M > 0$, and this set is not convex. Indeed, for example taking the convex combination of two Dirac distributions, both distributions have variance 0 but depending on where the Dirac distribution are located the variance of the convex combination is arbitrary. It also involves an optimization in both the first and second term of the KL because we consider the worst-case corruption in both the optimal arm distribution $\nu_\star$ and non-optimal arm distribution $\nu_i$. In this section, we do not solve these problems, but we propose lower bounds derived from the study of a specific class of heavy-tailed distributions on one hand (Lemma 3) and the study of a specific class of corrupted (but not heavy-tailed) distributions on the other hand (Lemma 4).

Using the fact that $\mathcal{K}_i(\nu_i, \mu(P_\star))$ is an infimum that is smaller than the $D_{\mathrm{KL}}$ for the choice $\nu = P_\star$, Lemma 2 induces the following weaker lower-bound:

$$\forall i \in \{1, \dots, k\}, \mu(\nu_i) \leq \mu_\star(\nu) \quad \Rightarrow \quad \liminf_{n \to \infty} \frac{\mathbb{E}_\nu[T_i(n)]}{\log(n)} \geq \frac{1}{D_{\mathrm{KL}}(\nu_i, P_\star)}. \tag{1}$$

Equation (1) shows that *it is sufficient to have an upper bound on the $D_{\mathrm{KL}}$-divergence of the reward distributions interacting with the policy to get a lower bound on the number of pulls of a suboptimal arm.*

In order to bound the $D_{\mathrm{KL}}$-divergence, we separately focus on two families of reward distributions, namely Student's distribution without corruption (Lemma 3) and corrupted Bernoulli distribution (Lemma 4), that reflect the hardness due to heavy-tails and corruptions, respectively. Applying Lemma 3 and Lemma 4 in Equation (1) yields the final regret lower bound in Theorem 1.

**Shifted Student's distribution without corruption.** To obtain a lower bound in the heavy-tailed case we use shifted Student distributions. Student distribution are well adapted because they exhibit a finite number of finite moment which makes them heavy-tailed, and we can easily change the mean of Student distribution by adding a shift without changing its shape parameter $d$. We denote by $\mathcal{T}_d$ the set of shifted Student distributions with $d$ degrees of freedom,

$$\mathcal{T}_d = \left\{ P \in \mathcal{P} : \exists \mu \in \mathbb{R} \ P \text{ has distribution } p : t \in \mathbb{R} \mapsto \frac{\Gamma(\frac{d+1}{2})}{\Gamma(d/2)\sqrt{d\pi}} \left( 1 + \frac{(t-\mu)^2}{d} \right)^{-\frac{d+1}{2}} \right\}.$$

**Lemma 3 (Control of KL-divergence for Heavy-tails)** *Let $P_1, P_2$ be two shifted Student distributions with $d > 1$ degrees of freedom with $\mathbb{E}_{P_1}[X] = 0$ and $\mathbb{E}_{P_2}[X] = \Delta > 0$. Then,*

$$D_{\mathrm{KL}}(P_1, P_2) \leq \begin{cases} \frac{3^{d-1}(d+1)^2\Delta^2}{5\sqrt{d}} & \text{if } \Delta \leq 1, \\ (d+1)\log(\Delta) + \log\left(3^d \frac{(d+1)^2}{5\sqrt{d}}\right) & \text{if } \Delta > 1. \end{cases} \tag{2}$$

**Corrupted Bernoulli distributions.** We denote $\mathcal{B}_p(\varepsilon) = \{(1-\varepsilon)P + \varepsilon H; H \sim \mathrm{Ber}(p') \text{ and } P \sim \mathrm{Ber}(p), p' \in [0,1]\}$ the corrupted neighborhood of the Bernoulli distribution $\mathrm{Ber}(p)$. Let $P_0 \in \mathrm{Ber}(p_0)$ and $P_1 \in \mathrm{Ber}(p_1)$ for some $p_0, p_1 \in (0,1)$ be two Bernoulli distributions. We corrupt both $P_0$ and $P_1$ with a proportion $\varepsilon > 0$ to get $Q_0 \in \mathcal{B}_{p_0}(\varepsilon)$ and $Q_1 \in \mathcal{B}_{p_1}(\varepsilon)$. We obtain Lemma 4 that illustrates three bounds on $D_{\mathrm{KL}}(Q_0, Q_1)$ as functions of the sub-optimality gap $\Delta \triangleq \mathbb{E}_{P_0}[X] - \mathbb{E}_{P_1}[X]$, variance $\sigma^2 \triangleq \mathrm{Var}_{P_0}(X) = \mathrm{Var}_{P_1}(X)$, and corruption proportion $\varepsilon$.

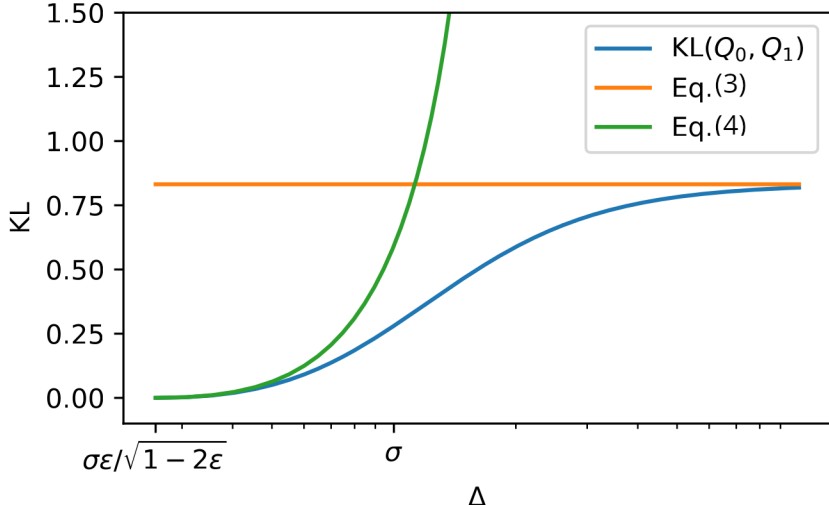

Figure 1: Visualizing the KL and the corresponding bounds in Lemma 4 for $\sigma = 1$ and $\varepsilon = 0.2$ ($x$ axis is in log scale).

**Lemma 4 (Control of KL-divergence for Corruptions)** *Let $P_0 \in \mathrm{Ber}(p_0)$ and $P_1 \in \mathrm{Ber}(p_1)$ be two Bernoulli probability distributions with means $p_0, p_1 \in (0, 1)$, such that $\Delta = \mathbb{E}_{P_0}[X] - \mathbb{E}_{P_1}[X] = p_0 - p_1$, $\sigma^2 = \mathrm{Var}_{P_0}(X) = \mathrm{Var}_{P_1}(X) > 0$, and $\Delta \geq \frac{2\sigma\varepsilon}{1-\varepsilon}$. Then, there exists $Q_0 \in \mathcal{B}_{p_0}(\varepsilon)$ and $Q_1 \in \mathcal{B}_{p_1}(\varepsilon)$, that have corrupted suboptimality gap given by $\overline{\Delta}_\varepsilon = \mathbb{E}_{Q_0}[X] - \mathbb{E}_{Q_1}[X] = \Delta(1-\varepsilon) - 2\varepsilon\sigma$, such that*

- **Uniform Bound.** *Without further assumptions on $\Delta$ and $\sigma$, we have*

$$D_{\mathrm{KL}}(Q_0, Q_1) \leq (1 - 2\varepsilon) \log\left(1 + \frac{1 - 2\varepsilon}{\varepsilon}\right). \tag{3}$$

- **High Distinguishability/Low Variance Regime.** *If $2\sigma \frac{\varepsilon}{\sqrt{1-2\varepsilon}} < \Delta < 2\sigma$, we get*

$$D_{\mathrm{KL}}(Q_0, Q_1) \leq \frac{\overline{\Delta}_\varepsilon}{2\sigma} \log\left(1 + \frac{\overline{\Delta}_\varepsilon}{2\sigma - \overline{\Delta}_\varepsilon}\right). \tag{4}$$

- **Low Distinguishability/High Variance Regime.** *If $\Delta \leq 2\sigma \frac{\varepsilon}{\sqrt{1-2\varepsilon}}$, there exists $\varepsilon' \leq \varepsilon$ and $Q_0' \in \mathcal{B}_{p_0}(\varepsilon')$, $Q_1' \in \mathcal{B}_{p_1}(\varepsilon')$ such that $D_{\mathrm{KL}}(Q_0', Q_1') = 0$.*

Note that the *corrupted* sub-optimality gap $\overline{\Delta}_\varepsilon$ should not be confused with the sub-optimality gap $\Delta$. Note also that due to the assumption on the variance in Lemma 4, we must have $p_0 = 1 - p_1 \geq 1/2$. The specific pair of distributions mentioned $Q_0, Q_1, Q_0', Q_1'$ can be found in the proof of the lemma, Section B.1.3.

**Consequences of Lemma 4.** We illustrate the bounds of Lemma 4 in Figure 1. The three upper bounds on the KL-divergence of corrupted Bernoullis provide us some insights regarding the impact of corruption.

1. *Three Regimes of Corruption:* We observe that, depending on $\Delta/\sigma$, we can categorize the corrupted environment in three categories. For $\Delta/\sigma \in [2, +\infty)$, we observe that the KL-divergence between corrupted distributions $Q_0$ and $Q_1$ is upper bounded by a function of only corruption proportion $\varepsilon$ and is independent of the uncorrupted distributions. Whereas for $\Delta/\sigma \in (2\varepsilon/\sqrt{1-2\varepsilon}, 2)$, the distinguishability of corrupted distributions depend on the distinguishibility of uncorrupted distributions and also the corruption level. We call this the High Distinguishability/Low Variance Regime. For $\Delta/\sigma \in [0, 2\varepsilon/\sqrt{1-2\varepsilon}]$, we observe that the KL-divergence can always go to zero. We refer to this setting as the Low Distinguishability/High Variance Regime.

2. *High Distinguishability/Low Variance Regime:* In Lemma 4, we observe that the effective gap to distinguish the optimal arm to the closest suboptimal arm that dictates hardness of a bandit instance has shifted from the uncorrupted gap $\Delta$ to a *corrupted suboptimality gap:* $\overline{\Delta}_\varepsilon \triangleq \Delta(1-\varepsilon) - 2\varepsilon\sigma$.

3. *Low Distinguishability/High Variance Regime:* We notice also that there is a limit for $\Delta$ below which the corruption can make the two distributions $Q_0$ and $Q_1$ indistinguishable, this is a general phenomenon in the setting of testing in corruption neighborhoods (Huber, 1965).

4. *Feasibility of the Bandits with Stochastic Corruption problem and $\overline{\Delta}_\varepsilon$:* In Lemma 4, we have assumed $\overline{\Delta}_\varepsilon$ to be positive. If $\overline{\Delta}_\varepsilon$ is negative or zero, i.e. $\frac{\Delta}{2\sigma} \leq \frac{\varepsilon}{1-\varepsilon}$, we cannot achieve better than linear regret in the corresponding Bandits with Stochastic Corruption problem. Lemma 4 additionally shows that we have to concede linear regret even when $\overline{\Delta}_\varepsilon$ is positive but $\frac{\Delta}{2\sigma} \leq \frac{\varepsilon}{\sqrt{1-2\varepsilon}}$.

**From KL Upper bounds to Regret Lower Bounds.** Substituting the results of Lemma 3 and 4 in Equation (1) yield the lower bounds on regret of any uniformly good policy in heavy-tailed and corrupted settings, where reward distributions either belong to the class of corrupted student distributions or the class of corrupted Bernoulli distributions, respectively. We denote

$$\mathfrak{D}_{\mathcal{T}_2}^{\otimes k} \triangleq \mathcal{T}_2 \otimes \cdots \otimes \mathcal{T}_2,$$

where $\mathcal{T}_2$ is the set of Student distributions with more than 2 degrees of freedoms. We also define

$$\mathfrak{D}_{\mathcal{B}(\varepsilon)}^{\otimes k} \triangleq \mathcal{B}(\varepsilon) \otimes \cdots \otimes \mathcal{B}(\varepsilon),$$

where $\mathcal{B}(\varepsilon) = \{(1-\varepsilon)P + \varepsilon H; H \sim \text{Ber}(p) \text{ and } P \sim \text{Ber}(p'), p, p' \in [0,1]\}$ is the set of corrupted Bernoulli distributions.

**Theorem 1 (Lower bound for heavy-tailed and corrupted bandit)** *Let $i$ be a suboptimal arm such that $\mathbb{E}_{P_i}[X] \leq \max_a \mathbb{E}_{P_a}[X]$ and denote $\Delta_i \triangleq \max_a \mathbb{E}_{P_a}[X] - \mathbb{E}_{P_i}[X]$ and $\overline{\Delta}_{i,\varepsilon} \triangleq \Delta_i(1-\varepsilon) - 2\varepsilon\sigma_i$, suppose $\overline{\Delta}_{i,\varepsilon} > 0$.*

***Student's distributions.*** *Suppose that the arms are pulled according to a policy that is uniformly good on $\mathfrak{D}_{\mathcal{T}_2}^{\otimes k}$. Then, for all $\nu \in \mathfrak{D}_{\mathcal{T}_2}^{\otimes k}$,*

$$\liminf_{n\to\infty} \frac{\mathbb{E}_\nu[T_i(n)]}{\log(n)} \geq \frac{\sigma_i^2}{51\Delta_i^2} \vee \frac{1}{4\log(\Delta_i/\sigma_i) + 22}. \tag{5}$$

***Corrupted Bernoulli distributions***: *Suppose that the arms are pulled according to a policy that is uniformly good on $\mathfrak{D}_{\mathcal{B}(\varepsilon)}^{\otimes k}$. Then, for all $\nu^\varepsilon \in \mathfrak{D}_{\mathcal{B}(\varepsilon)}^{\otimes k}$ such that $2\sigma_i \frac{\varepsilon}{\sqrt{1-2\varepsilon}} < \Delta_i < 2\sigma_i$, then*

$$\liminf_{n\to\infty} \frac{\mathbb{E}_{\nu^\varepsilon}[T_i(n)]}{\log(n)} \geq \frac{2\sigma_i}{\overline{\Delta}_{i,\varepsilon} \log\left(1 + \frac{\overline{\Delta}_{i,\varepsilon}}{2\sigma_i - \overline{\Delta}_{i,\varepsilon}}\right)}, \tag{6}$$

*and for $\Delta_i > 2\sigma_i$,*

$$\liminf_{n\to\infty} \frac{\mathbb{E}_{\nu^\varepsilon}[T_i(n)]}{\log(n)} \geq \frac{1}{(1-2\varepsilon)\log\left(\frac{1-\varepsilon}{\varepsilon}\right)}. \tag{7}$$

For brevity, the detailed proof is deferred to Appendix A.1.

**Small gap versus large gap regimes.** Due to the restriction in the family of distributions considered in Theorem 1, the lower bounds are not tight and may not exhibit the correct rate of convergence for all families of distributions. However, this theorem provides some insights about the difficulties that one may encounter in corrupted and heavy-tail bandits problems, including the logarithmic dependence on $n$.

In Theorem 1, if $\Delta_i$ is small, we see that in the heavy-tailed case (Student's distribution), we recover a term very similar to the lower bound when the arms are from a Gaussian distribution. Now, in the case where

$\Delta_i$ is large, the number of suboptimal pulls in the heavy-tail setting is $\Omega\left(1/\log\left(\frac{\Delta_i}{\sigma_i}\right)\right)$. This is the price to pay for heavy-tails.

If we are in the high distiguishability/low variance regime, i.e. $\frac{\overline{\Delta}_{i,\varepsilon}}{2\sigma_i} \in (\frac{\varepsilon}{\sqrt{1-2\varepsilon}}, 1)$, we recover a logarithmic lower bound which depends on a *corrupted gap between means* $\overline{\Delta}_{i,\varepsilon} = \Delta_i(1-\varepsilon) - 2\varepsilon\sigma_i$. Since the corrupted gap is always smaller than the true gap $\Delta_i$, this indicates that a corrupted bandit ($\varepsilon > 0$) must incur higher regret than a uncorrupted one ($\varepsilon = 0$). For $\varepsilon = 0$, this lower bound coincides with the lower bound for Gaussians with uncorrupted gap of means $\Delta_i$ and variance $\sigma_i^2$. On the other hand, if $\frac{\overline{\Delta}_{i,\varepsilon}}{2\sigma_i}$ is larger than 1, we observe that we can still achieve logarithmic regret but the hardness depends on only the corruption level $\varepsilon$, specifically $\frac{1}{(1-2\varepsilon)\log\left(\frac{1-\varepsilon}{\varepsilon}\right)}$.

## 5 Robust bandit algorithm: Huber's estimator and upper bound on the regret

In this section, we propose an UCB-type algorithm, namely `HuberUCB`, addressing the Bandits with Stochastic Corruption problem (Algorithm 2). This algorithm uses primarily a robust mean estimator called Huber's estimator (Section 5.1) and corresponding confidence bound to develop `HuberUCB` (Section 5.2). We further provide a theoretical analysis in Theorem 3 leading to upper bound on regret of `HuberUCB`. We observe that the proposed upper bound matches the lower bound in Theorem 1 under some settings.

### 5.1 Robust mean estimation and Huber's estimator

We begin with a presentation of the Huber's estimator of mean (Huber, 1964).

As we aim to design a UCB-type algorithm, the main focus is to obtain an empirical estimate of the mean rewards. Since the rewards are heavy-tailed and corrupted in this setting, we have to use a robust estimator of mean. We choose to use Huber's estimator (Huber, 1964), an M-estimator that is known for its robustness properties and have been extensively studied (e.g. the concentration properties (Catoni, 2012)).

Huber's estimator is an M-estimator, which means that it can be derived as a minimizer of some loss function. Given access to $n$ i.i.d. random variables $X_1^n \triangleq \{X_1, \ldots, X_n\}$, we define Huber's estimator as

$$\text{Hub}_\beta(X_1^n) \in \arg\min_{\theta \in \mathbb{R}} \sum_{i=1}^{n} \rho_\beta(X_i - \theta), \tag{8}$$

where $\rho_\beta$ is Huber's loss function with parameter $\beta > 0$. $\rho_\beta$ is a loss function that is quadratic near 0 and linear near infinity, with $\beta$ thresholding between the quadratic and linear behaviors.

In the rest of the paper, rather than using the aforementioned definition, we represent the Huber's estimator as a root of the following equation (Mathieu, 2022):

$$\sum_{i=1}^{n} \psi_\beta\left(X_i - \text{Hub}_\beta(X_1^n)\right) = 0. \tag{9}$$

Here, $\psi_\beta(x) \triangleq x\mathbf{1}\{|x| \le \beta\} + \beta\,\text{sign}(x)\mathbf{1}\{|x| > \beta\}$ is called the influence function. Though the representations in Equation (8) and (9) are equivalent, we prefer to use representation Equation (9) as we prove the properties of Huber's estimator using those of $\psi_\beta$.

$\beta$ plays the role of a scaling parameter. Depending on $\beta$, Huber's estimator exhibits a trade-off between the efficiency of the minimizer of the square loss, i.e. the empirical mean, and the robustness of the minimizer of the absolute loss, i.e. the empirical median.

### 5.2 Concentration of Huber's estimator in corrupted setting

Let use denote the true Huber mean for a distribution $P$ as $\text{Hub}_\beta(P)$. This means that, for a random variable $Y$ with law $P$, $\text{Hub}_\beta(P)$ satisfies $\mathbb{E}[\psi_\beta(Y - \text{Hub}_\beta(P))] = 0$.

We now state our first key result on the concentration of Huber's estimator around $\mathrm{Hub}_\beta(P)$ in a corrupted and heavy-tailed setting.

**Theorem 2 (Concentration of Empirical Huber's estimator)** *Suppose that $X_1, \ldots, X_n$ are i.i.d. with law $(1 - \varepsilon)P + \varepsilon H$ for some $P, H \in \mathcal{P}$ and proportion of outliers $\varepsilon \in (0, 1/2)$, and $P$ has a finite variance $\sigma^2$. Then, with probability larger than $1 - 5\delta$,*

$$|\mathrm{Hub}_\beta(X_1^n) - \mathrm{Hub}_\beta(P)| \leq \frac{\sigma\sqrt{\frac{2\ln(1/\delta)}{n}} + \beta\frac{\ln(1/\delta)}{3n} + 2\beta\overline{\varepsilon}\sqrt{\frac{\ln(1/\delta)}{n}} + 2\beta\varepsilon}{\left(p - \sqrt{\frac{\ln(1/\delta)}{2n}} - \varepsilon\right)_+}.$$

*Here, $p = \mathbb{P}_P(|Y - \mathbb{E}_P[Y]| \leq \beta/2)$ with $p > 5\varepsilon$, $\beta > 4\sigma$, $\overline{\varepsilon} = \sqrt{\frac{(1-2\varepsilon)}{\log\left(\frac{1-\varepsilon}{\varepsilon}\right)}}$, and $\delta \geq \exp\left(-n\frac{128(p-5\varepsilon)^2}{49\left(1+2\overline{\varepsilon}\sqrt{2}\right)^2}\right)$.*

Theorem 2 gives us the concentration of $\mathrm{Hub}_\beta(X_1^n)$ around $\mathrm{Hub}_\beta(P)$, i.e. the Huber functional of the *inlier* distribution $P$. This theorem allows us to construct a UCB-type algorithm to solve the Bandits with Stochastic Corruption.

For convenience of notation, hereafter, we denote the rate of convergence of $\mathrm{Hub}_\beta(X_1^n)$ to $\mathrm{Hub}_\beta(P)$ as

$$r_n(\delta) \triangleq \frac{\sigma\sqrt{\frac{2\ln(1/\delta)}{n}} + \beta\frac{\ln(1/\delta)}{3n} + 2\beta\overline{\varepsilon}\sqrt{\frac{\ln(1/\delta)}{n}} + 2\beta\varepsilon}{\left(p - \sqrt{\frac{\ln(1/\delta)}{2n}} - \varepsilon\right)_+}. \tag{10}$$

**Discussion.** Now, we provide a brief discussion on the implications of Theorem 2.

*1. Value of $p$:* For most laws that exhibit concentration properties, the constant $p$ is close to 1 as $\beta \geq 4\sigma$. One might also use Markov inequality to lower bound $p$, depending on the number of finite moments $P$ has. Bounding $p$ then becomes a trade-off on the value of $\beta$, where large values of $\beta$ implies that $p$ is close to 1. But larger $\beta$ also leads to a less robust estimator, since the error bound in Theorem 2 increases with $\beta$.

*2. Tightness of constants:* If there are no outliers ($\varepsilon = 0$), the optimal rate of convergence in such a setting is at least of order $\sigma\sqrt{2\ln(1/\delta)/n}$ due to the central limit theorem. Theorem 2 shows that we are very close to attaining this optimal constant in the leading $1/\sqrt{n}$ term. This result for Huber's estimator echoes the one presented in Catoni (2012).

*3. Value of $\beta$:* $\beta$ is a parameter that achieve a trade-off between accuracy in the light-tailed uncorrupted setting and robustness. For our result, $\beta$ must be at least of the order of $4\sigma$. We provide a detailed discussion on the choice of $\beta$ in Section 5.4.

*4. Restriction on the values of $\delta$:* In Theorem 2, $\delta$ must be at least of order $e^{-n}$. This restriction may seem arbitrary but it is in fact unavoidable as shown in Theorem 4.3 of Devroye et al. (2016). This is a limitation of robust mean estimation that enforces our algorithm to perform a forced exploration in the beginning.

*5. Restriction on the values of $\varepsilon$:* In Theorem 2, $\varepsilon$ can be at most $p/5$, which implies that it is smaller than $1/5$. This restriction is common in robustness literature. In particular, in Kapoor et al. (2019), $\varepsilon$ is supposed smaller than $\Delta/\sigma$. In robustness literature, Lecué & Lerasle (2020) and Dalalyan & Thompson (2019) assumed that $\varepsilon \leq 1/768$ and $1/400$ respectively. In contrast, our analysis can handle $\varepsilon$ up to 0.2, which is significantly higher than the existing restrictions.

**Bias of Huber's Estimate.** If $P$ is symmetric, we have $\mathrm{Hub}_\beta(P) = \mathbb{E}[X]$. When $P$ is non-symmetric, we need to control the distance of the Huber's estimate from the true mean, i.e. $|\mathrm{Hub}_\beta(P) - \mathbb{E}[X]|$. We call it the bias of Huber's estimate. We need to bound this bias to get a concentration of the empirical Huber's estimate $\mathrm{Hub}_\beta(X_1^n)$ around the true mean $\mathbb{E}[X]$. We control the bias using the following lemma, which is a direct consequence of Lemma 4 from Mathieu (2022).

**Lemma 5 (Bias of Huber's estimator)** *Let $Y$ be a random variable with $\mathbb{E}[|Y|^q] < \infty$ for $q \geq 2$ and suppose that $\beta^2 \geq 9\mathrm{Var}(Y)$. Then*

$$|\mathbb{E}[Y] - \mathrm{Hub}_\beta(P)| \leq \frac{2\mathbb{E}[|Y - \mathbb{E}[Y]|^q]}{(q-1)\beta^{q-1}}.$$

Using Lemma 5 and Theorem 2, we can control the deviations of $\mathrm{Hub}_\beta(X_1^n)$ from $\mathbb{E}[X]$. This allows us to formulate an index-based algorithm (UCB-type algorithm) for corrupted Bandits. We present this algorithm in Section 5.3.

### 5.3 `HuberUCB`: Algorithm and regret bound

In this section, we describe a robust, UCB-type algorithm called `HuberUCB`. We denote $\mu_i$ as the mean of arm $i$ and its variance as $\sigma_i^2$. We assume that we know the variances of the reward distributions, i.e. $\{\sigma_i^2\}_{i=1}^k$, and hence, we define by construction $M \triangleq \max_i \sigma_i^2$. We refer to Section 5.4 for a discussion on the choice of the parameters when the reward distributions are unknown.

`HuberUCB`: **The algorithm.** In order to deploy the Huber's estimator in the multi-armed bandits setting, we need to estimate the mean of the rewards of each arm separately. We do that by defining a parameter $\beta_i$ for each arm and estimating separately each $\mu_i$ using

$$\mathrm{Hub}_{i,s} = \mathrm{Hub}_{\beta_i}\left(X_t, \quad 1 \leq t \leq s \quad \text{such that} \quad A_t = i,\right).$$

Now, at each step $t$, we define a confidence bound for arm $i$ with $s$ number of pulls as

$$B_i(s,t) \triangleq \begin{cases} r_s(1/t^2) + b_i & \text{if } s \geq s_{lim}(t) \\ \infty & \text{if } s < s_{lim}(t) \end{cases}, \tag{11}$$

where $r_s(1/t^2)$ is defined by Equation (10), $s_{lim}(t) = \log(t)\frac{98}{128(p-5\varepsilon)^2}\left(1 + 2\sqrt{2}\left(\overline{\varepsilon} \vee \frac{9}{14\sqrt{2}}\right)\right)^2, \overline{\varepsilon} = \sqrt{\frac{(1-2\varepsilon)}{\log\left(\frac{1-\varepsilon}{\varepsilon}\right)}}$, and $b_i$ is a bound on the bias $|\mathbb{E}[X] - \mathrm{Hub}_{\beta_i}(P_i)|$. Hence $b_i$ can be set to zero if $P_i$ is known to be symmetric and controlled by Lemma 5 otherwise. Here, we assign $b_i = 2\sigma_i^2/\beta_i$ as a conservative choice by imposing $q = 2$, i.e. finite second moment, in Lemma 5.

Now, we propose `HuberUCB` that selects an arm $a_t$ at step $t$ based on the index

$$I_i^{\texttt{HuberUCB}}(t) = \mathrm{Hub}_{i,T_i(t-1)} + B_i(T_i(t-1),t). \tag{12}$$

The index of `HuberUCB` together with the confidence bound defined in Equation (11) dictates that if an arm is less explored, i.e. $T_i(t-1) < s_{lim}(t)$, we choose that arm, and if multiple arms satisfy this, we break the tie randomly. As $t$ grows and for all the arms $T_i(t-1) \geq s_{lim}(t)$ is satisfied, we choose the arms according to the adaptive bonus. Thus, `HuberUCB` induces an initial forced exploration to obtain confident-enough robust estimates, followed by a time-adaptive selection of arms. We present a pseudocode of `HuberUCB` in Algorithm 2. We discuss the choices of the hyperparameters and the computational details in Section 5.4.

---
**Algorithm 2** `HuberUCB`
---
**Require:** Parameter $\varepsilon \in [0, 1/2)$ and $\beta_i > 4\sigma_i$ for all $i \leq K$
 1: **for** $t = 1, \ldots, n$ **do**
 2:     Compute index $I_i^{\texttt{HuberUCB}}(t)$ (Equation (12)) for $i \in \{1, \ldots, k\}$ using $X_1, \ldots, X_{t-1}$.
 3:     Choose arm $a_t \in \arg\max_i I_i(t)$.
 4:     Observe a reward $X_t$.
 5: **end for**

---

**Regret Analysis.** Now, we provide a regret upper bound for `HuberUCB`.

**Theorem 3 (Upper Bound on number of pulls of suboptimal arms with `HuberUCB`)** *Let us consider a set of $k$ reward distributions $\{P_i\}_{i=1}^k$ with known and finite variances $\{\sigma_i^2\}_{i=1}^k$, i.e. for all $i \in \{1, \ldots, k\}$, $P_i \in \mathcal{P}_{[2]}(M)$ such that $M \triangleq \max\limits_{i \in \{1, \ldots, k\}} \sigma_i^2$. Let us also consider some $\beta_i \geq 4\sigma_i$ and $p \triangleq \inf_{1 \leq i \leq k} \mathbb{P}_{P_i}(|X - \mathbb{E}_{P_i}[X]| \leq \beta_i/2)$ such that $p > 5\varepsilon$ and $\varepsilon < 1/5$. We denote $\widetilde{\Delta}_{i,\varepsilon} \triangleq (\Delta_i - 2b_i)(p - \varepsilon) - 8\beta_i\varepsilon$, which we assume positive and $\sqrt{\frac{(1-2\varepsilon)}{\log(\frac{1-\varepsilon}{\varepsilon})}} \leq \overline{\varepsilon}$.*

- *If $\widetilde{\Delta}_{i,\varepsilon} > 12\frac{\sigma_i^2}{\beta_i}\left(\sqrt{2} + 2\frac{\beta_i}{\sigma_i}\overline{\varepsilon}\right)^2$, then `HuberUCB` pulls in expectation arm $i$ at most*

$$\mathbb{E}[T_i(n)] \leq \log(n) \max\left(\frac{32\beta_i}{3\widetilde{\Delta}_{i,\varepsilon}}, \frac{4}{(p-5\varepsilon)^2}\left(1 + 2\sqrt{2}\left(\overline{\varepsilon} \vee \frac{9}{14\sqrt{2}}\right)\right)^2\right) + 10(\log(n)+1)$$

- *If $\widetilde{\Delta}_{i,\varepsilon} \leq 12\frac{\sigma_i^2}{\beta_i}\left(\sqrt{2} + 2\frac{\beta_i}{\sigma_i}\overline{\varepsilon}\right)^2$, then `HuberUCB` pulls in expectation arm $i$ at most*

$$\mathbb{E}[T_i(n)] \leq \log(n) \max\left(\frac{50\sigma_i^2}{9\widetilde{\Delta}_{i,\varepsilon}^2}\left(\sqrt{2} + 2\frac{\beta_i}{\sigma_i}\overline{\varepsilon}\right)^2, \frac{4}{(p-5\varepsilon)^2}\left(1 + 2\sqrt{2}\left(\overline{\varepsilon} \vee \frac{9}{14\sqrt{2}}\right)\right)^2\right) + 10(\log(n)+1).$$

Using Theorem 3 and Lemma 1, a bound on the corrupted regret of `HuberUCB` follows immediately.

We now state a simplified version of Theorem 3 with worse but explicit constants for easier comprehension. Let us fix $\beta_i^2 = 16\sigma_i^2$ and $\varepsilon \leq 1/10$ such that $\overline{\varepsilon} = 4/(5\sqrt{\ln(9)}) \simeq 0.54$, and $p \geq 1 - \frac{4\sigma_i^2}{\beta_i^2} \geq \frac{3}{4} \geq 5\varepsilon + \frac{1}{4}$. Now, if we further assume that $P_i$ symmetric leading to $b_i = 0$, it yields the following upper bounds.

**Corollary 1 (Simplified version of Theorem 3)** *Suppose that for all $i$, $P_i$ is a symmetric distribution with finite variance $\sigma_i^2$. Let also denote $\widetilde{\Delta}_{i,\varepsilon} \triangleq \Delta_i(p - \varepsilon) - 32\sigma_i\varepsilon$ which is assumed to be positive and let $\varepsilon < 1/10$.*

- *If $\widetilde{\Delta}_{i,\varepsilon} > 6\sigma_i\left(1 + 4\sqrt{2}\overline{\varepsilon}\right)^2$, then `HuberUCB` pulls in expectation arm $i$ at most*

$$\mathbb{E}[T_i(n)] \leq 43\log(n)\max\left(\frac{\sigma_i}{\widetilde{\Delta}_{i,\varepsilon}}, 10\right) + 10(\log(n) + 1).$$

- *If $\widetilde{\Delta}_{i,\varepsilon} \leq 6\sigma_i\left(1 + 4\sqrt{2}\overline{\varepsilon}\right)^2$, then `HuberUCB` pulls in expectation arm $i$ at most*

$$\mathbb{E}[T_i(n)] \leq 23\log(n)\max\left(\frac{\sigma_i^2}{\widetilde{\Delta}_{i,\varepsilon}^2}\left(1 + 32\overline{\varepsilon}^2\right), 18\right) + 10(\log(n) + 1).$$

Remark that in this corollary, we replaced some occurrences of $\overline{\varepsilon}$ by its upper bound, which is also an upper bound on $\varepsilon$. Thus, the presented result is loose up to constants but lend itself to easier comprehension.

**Discussions on the Upper Bound.** Here, we discuss how this proposed upper bound of `HuberUCB` matches and mismatches with the lower bounds in Theorem 1.

1. *Order-optimality of Upper Bound.* `HuberUCB` achieves the logarithmic regret prescribed by the lower bound (Theorem 1) plus some additive error due to the fact that this is a UCB-type algorithm. Thus, `HuberUCB` is order optimal with respect to $n$.

2. *Two Regimes of Upper Bound.* When $\Delta_i$ is small compared to $\sigma_i$, we obtain an upper bound $\mathbb{E}[T_i(n)] \underset{n \to \infty}{=} \mathcal{O}\left(\log(n)\left(\frac{\sigma_i^2}{\widetilde{\Delta}_{i,\varepsilon}^2}\overline{\varepsilon}^2\right)\right)$ from Corollary 1. $\overline{\varepsilon}^2$ is of the same order of magnitude as Equation (7) because we take $\varepsilon$ strictly smaller than $1/2$. $\overline{\varepsilon}^2$ acts as an indicator of the corruption level. The term $\frac{\sigma_i^2}{\widetilde{\Delta}_{i,\varepsilon}^2}$ indicates

the hardness due to the corrupted gaps $\widetilde{\Delta}_{i,\varepsilon}$ and echoes the hardness term $\frac{\sigma_i^2}{\Delta_i^2}$ that appears in regret upper bound of UCB for uncorrupted bandits. The hardness term $\frac{\sigma_i^2}{\Delta_{i,\varepsilon}^2}$ also appears in the corrupted lower bound (Equation (6)) as well as the heavy-tailed lower bound (Equation (5)) for $\Delta_i \ll \sigma_i{}^2$.

On the other hand, if $\Delta_i$ is larger than $\sigma_i$, we get that $\mathbb{E}[T_i(n)] = O\left(\log(n)\left(\frac{\sigma_i}{\Delta_{i,\varepsilon}} \vee \bar{\varepsilon}^2 \vee 1\right)\right)$. This upper bound reflects the lower bound in Equation (7) that holds for $\Delta_i > 2\sigma_i$. This reinstates the fact that for large enough suboptimality gaps, the regret of `HuberUCB` depends solely on the corruption level than the suboptimality gap.

3. *Deviation from the Lower Bound.* The two regimes defined in the upper bound does not follow the exact distinctions made in the lower bounds. We observe that in the upper bound, the distinction between regimes depend on a corrupted suboptimality gap $\widetilde{\Delta}_{i,\varepsilon} \triangleq \Delta_i (p - \varepsilon) - 32\sigma_i\varepsilon$, while the lower bound depends on the corrupted suboptimality gap $\overline{\Delta}_{i,\varepsilon} \triangleq \Delta_i (1 - \varepsilon) - 2\sigma_i\varepsilon$. This difference in constants hinder the hardness regimes and corresponding constants in upper and lower bounds to match for all $\Delta_i, \sigma_i,$ and $\varepsilon$. This deviation also comes from the fact that the lower bounds proposed in Theorem 1 consider effects of heavy-tails and corruptions separately, while the upper bound of `HuberUCB` consider them in a coupled manner.

Additionally, we observe that regret of `HuberUCB` is suboptimal due to the constant additive error, which appears due to the initial forced exploration of `HuberUCB` up to $s_{lim}(t)$. Our concentration bounds and corresponding regret analysis shows that this forced exploration phase is unavoidable in order to be able to handle the case $\Delta_i \leq \sigma_i$ with `HuberUCB`. Removing this discrepancy between the lower and upper bounds would constitute an interesting future work.

## 5.4 Computational Details

Here, we discuss the three hyperparameters that `HuberUCB` depends on and also its computational cost.

*Choice of $\sigma$ and $\varepsilon$.* In Theorem 3, we assume to know the $\sigma$ and $\varepsilon$. In practice, these are unknown and we estimate $\sigma^2$ with a robust estimator of the variance, such as the median absolute deviation. In contrast, estimating $\varepsilon$ is hard. There exists some heuristics, for example using the proportion of point larger than 1.5 times the inter-quartile range or using more complex algorithms like Isolation Forest algorithm but these methods work in general using the hypothesis that outliers are in some way points that are located outside of "the bulk of the data" which conflicts with the fact that we don't suppose anything on the outliers. Moreover even though there are heuristics, the problem of finding what constitute "the bulk of the data" is closely linked to problems such as finding a "Robust minimum volume ellipsoid" which is NP-hard in general (Mittal & Hanasusanto, 2022). We refer to Appendix C.1 for an ablation study on the choice of $\varepsilon$.

*Choice of $\beta_i$.* Ideally, $\beta_i$ should be larger than $4\sigma_i$. We recommend using an estimator of $\sigma_i$ to estimate a good value of $\beta_i$. The choice of $\beta_i$ reflects the difference between heavy-tailed bandits and corrupted bandits. When the data are heavy-tailed but not corrupted, Catoni (2012) shows that $\beta_i \simeq \sigma_i\sqrt{n}$ is a good choice for the scaling parameter. However, this choice is not robust to outliers and yields a linear regret in our setup (see Section 7) and a trade-off between Heavy-tailed and corrupted setting would dictate $\beta_i \simeq \sigma\sqrt{n} \wedge \varepsilon^{-1/2}$ (see Proposition 2 in Mathieu (2022)). In Appendix C.1, we present an ablation study on the choice of $\varepsilon$.

*Computational Cost.* Huber's estimator has linear complexity due to the involved Iterated Re-weighting Least Squares algorithm, which is not sequential. We have to do this at every iteration, which leads `HuberUCB` to have a quadratic time complexity. This is the computational cost of using a robust mean estimator, i.e. the Huber's estimator.

---

[2]We observe that the lower bound in Equation (5) depends on $\frac{\sigma_i^2}{\Delta_{i,\varepsilon}^2}$ for $\Delta_i \ll \sigma_i$, since the first order approximation of $\log(1 + x)$ is $x$ as $x \to 0$.

## 6    `SeqHuberUCB`: A Faster Robust Bandit Algorithm

In this section, we present a sequential approximation of the Huber's estimator, and we leverage it further to create a robust bandit algorithm with linear-time complexity algorithm. Here, we describe the algorithm (`SeqHuberUCB`) and its theoretical properties.

**A sequential approximation of Huber's estimator.**    The central idea is to compute the Huber's estimator using the full historical data only in logarithmic number of steps than at every step, and in between two of these re-computations, update the estimator using only the samples observed at that step. This allows us to propose a sequential approximation of Huber's estimator, i.e. $\mathrm{SeqHub}_t$, with lower computational complexity.

By fixing the update step $P_2(t) = 2^{\left\lfloor \frac{\log(t)}{\log(2)} \right\rfloor}$ before a given step $t > 0$, we define the estimator $\mathrm{SeqHub}_t$ by $\mathrm{SeqHub}_0 = 0$ and

$$\mathrm{SeqHub}_t = \begin{cases} H_t & \text{if } t = P_2(t), \\ H_t + \frac{\sum_{i=P_2(t)}^{t} \psi(X_i - H_t)}{\sum_{i=1}^{t} \psi'(X_i - H_t)} & \text{otherwise.} \end{cases} \tag{13}$$

Here, $H_t \triangleq \mathrm{Hub}(X_1^{P_2(t)})$ and $\psi$ is the influence function defined in Equation (9). $\mathrm{SeqHub}_t$ can be conceptualized as a first order Taylor approximation of $\mathrm{Hub}(X_1^t)$ around $\mathrm{Hub}(X_1^{P_2(t)})$.

One might argue that $\mathrm{SeqHub}_t$ is not fully sequential rather a phased estimator as we still recompute the Huber's estimator following a geometric schedule. Thus, we still need to keep all the data in memory, leading to linear space complexity as the non-sequential Huber's estimator. But it features the good property of having a linear time complexity when computed using the prescribed geometric schedule. This implies that the `SeqHuberUCB` algorithm leveraging the sequential Huber's estimator achieves a linear time complexity.

**Concentration Properties of** $\mathrm{SeqHub}$**.**  Now, in order to propose `SeqHuberUCB` we first aim to derive the rate of convergence of $\mathrm{SeqHub}_t$ towards the true Huber's mean $\mathrm{Hub}(P)$.

**Theorem 4** *If the assumptions of Theorem 2 hold true, with probability larger than $1 - 14\delta$, we have*

$$|\mathrm{SeqHub}_t - \mathrm{Hub}(P)| \leq r_t(\delta) + \left( \frac{1}{p - \sqrt{\frac{\log(1/\delta)}{2t}} - \varepsilon} - 1 \right) r_{P_2(t)}(\delta) \tag{14}$$

*for any $t > 0$, and $\delta \geq \exp\left( -P_2(t) \frac{128(p-5\varepsilon)^2}{49\left(1+2\overline{\varepsilon}\sqrt{2}\right)^2} \right)$. Here, $r_t(\delta)$ is defined as in Equation (10).*

We observe that the confidence bound of $\mathrm{SeqHub}_t$ includes the confidence bound of $\mathrm{Hub}_t$, i.e. $r_t(\delta)$, and an additive term proportional to $r_{P_2(t)}(\delta)$. Since $r_{P_2(t)}(\delta) \geq r_t(\delta)$ for $t \geq P_2(t)$, we can show that $|\mathrm{SeqHub}_t - \mathrm{Hub}(P)| \leq \left( p - \sqrt{\frac{\log(1/\delta)}{2t}} - \varepsilon \right)^{-1} r_{P_2(t)}(\delta)$. Thus, we obtain larger confidence bounds for $\mathrm{SeqHub}$ than that of $\mathrm{Hub}$, and they differ approximately by a multiplicative constant $(p - \varepsilon)^{-1}$ as $t \to \infty$.

**`SeqHuberUCB`: The algorithm.**    Now, we plug-in the sequential Huber's estimator, $\mathrm{SeqHub}$, and the corresponding confidence bound (Equation (14)), instead of the Huber's estimator and the corresponding confidence bound in the `HuberUCB` algorithm. This allows us to construct the `SeqHuberUCB` algorithm that we present hereafter.

Specifically, we define the index of `SeqHuberUCB` as

$$I_i^{\mathtt{SeqHuberUCB}}(t) = \mathrm{SeqHub}_{i, T_i(t-1)} + B_i^{\mathtt{SeqHuberUCB}}(T_i(t-1), t). \tag{15}$$

where

$$\mathrm{SeqHub}_{i,s} = \mathrm{SeqHub}\left( X_t, \quad 1 \leq t \leq s \quad \text{such that} \quad A_t = i, \right),$$

and a confidence bound for arm $i$ with $s$ number of pulls is

$$B_i^{\texttt{SeqHuberUCB}}(s,t) \triangleq \begin{cases} r_s(1/t^2) + \left( \dfrac{1}{p - \sqrt{\frac{\log(1/\delta)}{2s}} - \varepsilon} - 1 \right) r_{P_2(s)}(1/t^2) + b_i & \text{if } P_2(s) \geq s_{lim}(t) \\ \infty & \text{if } P_2(s) < s_{lim}(t). \end{cases}$$

Here, $s_{lim}(t)$, $\overline{\varepsilon}$ and $b_i$ are same as defined for $\texttt{HuberUCB}$.

Similar to Corollary 1, we now present a simplified regret upper bound for $\texttt{SeqHuberUCB}$. Retaining the setting of Corollary 1, we assume that $\beta_i^2 = 16\sigma_i^2$, $\varepsilon \leq 1/10$ implying $\overline{\varepsilon} = 4/(5\sqrt{\ln(9)}) \simeq 0.54$, $p \geq 1 - \frac{4\sigma_i^2}{\beta_i^2} \geq \frac{3}{4} \geq 5\varepsilon + \frac{1}{4}$, and $P_i$ symmetric so that $b_i = 0$. Further simplifying the constants yields the following regret upper bound for $\texttt{SeqHuberUCB}$.

**Lemma 6 (Simplified Upper Bound on Regret of $\texttt{SeqHuberUCB}$)** *Suppose that for all $i$, $P_i$ is a distribution with finite variance $\sigma_i^2$. Let us also denote $\widetilde{\Delta}_{i,\varepsilon} = \Delta_i\,(p - \varepsilon) - 32\sigma_i\varepsilon$,*

- *If $\widetilde{\Delta}_{i,\varepsilon} > 18\sigma_i\left(1 + 4\sqrt{2}\overline{\varepsilon}\right)^2$, then*

$$\mathbb{E}[T_i(n)] \leq 128 \log(n) \max\left( \frac{\sigma_i}{\widetilde{\Delta}_{i,\varepsilon}}, 2 \right) + 28(\log(n) + 1).$$

- *If $\widetilde{\Delta}_{i,\varepsilon} \leq 18\sigma_i\left(1 + 4\sqrt{2}\overline{\varepsilon}\right)^2$, then*

$$\mathbb{E}[T_i(n)] \leq 80 \log(n) \max\left( \frac{\sigma_i^2}{\widetilde{\Delta}_{i,\varepsilon}^2}\left(1 + 32\overline{\varepsilon}^2\right), 3 \right) + 28(\log(n) + 1).$$

**Comparison between Regrets of $\texttt{HuberUCB}$ and $\texttt{SeqHuberUCB}$.** Lemma 6 yields similar regret bounds for $\texttt{SeqHuberUCB}$ as the ones obtained for $\texttt{HuberUCB}$ in Corollary 1. We observe that the regrets of these two algorithms only differ in $n$-independent constants. Specifically, regret of $\texttt{SeqHuberUCB}$ can be approximately $3-4$ times higher than that of $\texttt{HuberUCB}$. For simplicity of exposition, we present approximate constants in our results. A more careful analysis might yield more fine-tuned constants. Theorem 4 and experimental results (Figure 2) indicate that it is possible to have very close performances with $\texttt{SeqHuberUCB}$ and $\texttt{HuberUCB}$.

## 7 Experimental Evaluation

In this section, we assess the experimental efficiency of $\texttt{HuberUCB}$ and $\texttt{SeqHuberUCB}$ by plotting the empirical regret. Contrary to the uncorrupted case, we cannot really estimate the corrupted regret in (Corrupted regret) only using the observed rewards. Instead, we use the true uncorrupted gaps that we know because we are in a simulated environment, and we estimate the corrupted regret $R_n$ using $\sum_{i=1}^{k} \Delta_i \widehat{T_i}(n)$, where $\widehat{T_i}(n) = \frac{1}{M} \sum_{m=1}^{M} (T_i(n))_m$ is a Monte-Carlo estimation of $\mathbb{E}_{\nu^\varepsilon}[T_i(n)]$ over $M$ experiments. We use rlberry library (Domingues et al., 2021) and Python3 for the experiments. We run the experiments on an 8 core Intel(R) Core(TM) i7-8665U CPU@1.90GHz. For each algorithm, we perform each experiment 100 times to get a Monte-Carlo estimate of regret.

**Comparison with Bandit Algorithms for Heavy-tailed and Adversarial Settings.** To the best of our knowledge, there is no existing bandit algorithm for handling unbounded stochastic corruption prior to this work. Hence, we focus on comparing ourselves to the closest settings, i.e. bandits in heavy-tailed setting and adversarial bandit algorithms. We empirically and competitively study five different algorithms: $\texttt{HuberUCB}$, $\texttt{SeqHuberUCB}$, two RobustUCB algorithms with Catoni-Huber estimator and Median of Means (MOM) (Bubeck et al., 2013). In particular, we compare to algorithms assuming bounded *centered* moments and not bounded raw moment such as Truncated Mean from Bubeck et al. (2013), and $\text{KL}_{\inf}$-UCB from Agrawal et al. (2021). See also Appendix C for further experimental results.

$\texttt{HuberUCB}$ is closely related to the RobustUCB with Catoni Huber estimator, which also uses Huber's estimator but with another set of parameters and confidence intervals. The RobustUCB algorithms are tuned

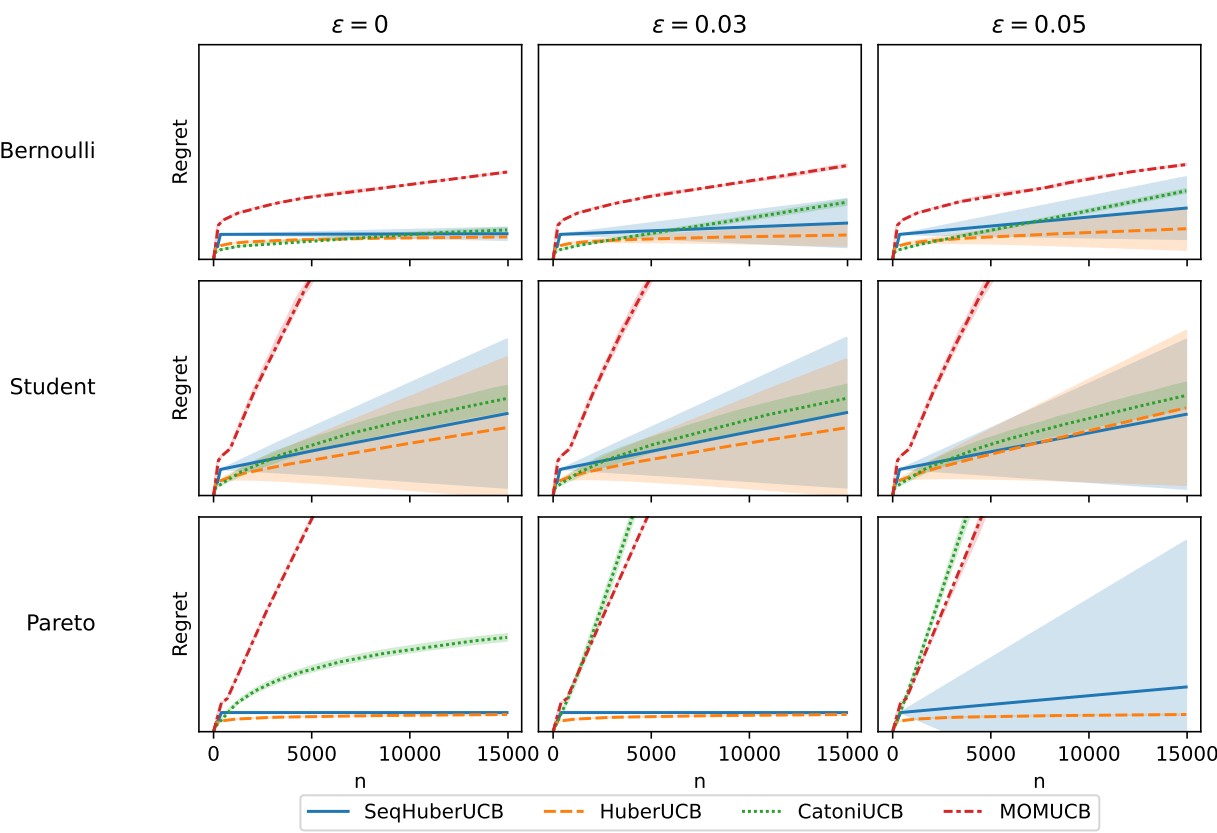

Figure 2: Cumulative regret plot of the algorithms on a corrupted Bernoulli (above), Student's (middle) and Pareto (below) reward distributions with various corruption levels $\varepsilon$. Lower corrupted regret indicates better performance for an algorithm.

for uncorrupted heavy-tails. Hence, they incur linear regret in a corrupted setting. This is reflected in the experiments. *We also improve upon Bubeck et al. (2013) as we can handle arm-dependent variances.*

**Corrupted Bernoulli setting:** In Figure 2 (above), we study a 3-armed bandits with corrupted Bernoulli distributions with means $0.1, 0.97, 0.99$. The corruption applied to this bandit problem are Bernoulli distributions with means $0.999, 0.999, 0.001$, respectively. For `HuberUCB` and `SeqHuberUCB`, we choose to use $\beta_i = 0.1\sigma_i$, which seems to work better despite the theory presented before. We plot the mean plus/minus the standard error of the result in Figure 2. We do that for the three corruption proportions $\varepsilon$ equal to $0\%$, $3\%$ and $5\%$. We notice that there is a short linear regret phase at the beginning due to the forced exploration performed by the algorithms. Followed by that, `HuberUCB` and `SeqHuberUCB` incur logarithmic regret. On the other hand, Catoni Huber Agent and MOM Agent incur logarithmic regret only in the uncorrupted setting. When the data are corrupted, i.e. $\varepsilon > 0$, their regret grow linearly.

**Corrupted Student setting:** In Figure 2 (middle), we study a 3-armed bandits with corrupted Student's distributions with 3 degrees of freedom (finite second moment) and with means $0.1, 0.95, 1$. The corruption applied to this bandit problem are Gaussians with variance $1$, and means $100, 100, -1000$ respectively. For `HuberUCB` and `SeqHuberUCB`, we choose to use $\beta_i = \sigma_i$. The results echo the observations for the Bernoulli case except that the corruption is more drastic and affect the performance even more.

**Corrupted Pareto setting:** In Figure 2 (bottom), we illustrate the results for a 3-armed bandits with corrupted Pareto distributions having shape parameters $3, 3, 2.1$ (i.e. they have finite second moments), and scale parameters $0.1, 0.2, 0.3$ respectively. Thus, the corresponding means are $0.15, 0.3$ and $0.57$ and the standard deviations are $0.09, 0.17, 1.25$, respectively. The corruption applied to this bandit problem are

Gaussians with variance 1, and centered at $100, 100, -1000$ respectively. For `HuberUCB` and `SeqHuberUCB`, we choose to use $\beta = 1.5\sigma_i$ and we also bound the bias $b_i$ by $\sigma_i^2/\beta_i$. The results echo the observations for the Student's distributions.

Thus, we conclude that `HuberUCB` incur the lowest regret among the competing algorithms in the Bandits with Stochastic Corruption setting, specially for higher corruption levels $\varepsilon$. Also, performances of `SeqHuberUCB` and `HuberUCB` are very close, except for the Pareto distributions with high corruption level.

## 8 Conclusion

In this paper, we study the setting of Bandits with Stochastic Corruption that encompasses both the heavy-tailed rewards with bounded variance and unbounded corruptions in rewards. In this setting, we prove lower bounds on the regret that shows the heavy-tailed bandits and corrupted bandits are strictly harder than the usual sub-Gaussian bandits. Specifically, in this setting, the hardness depends on the suboptimality gap/variance regimes. If the suboptimality gap is small, the hardness is dictated by $\sigma_i^2/\overline{\Delta}_{i,\varepsilon}^2$. Here, $\overline{\Delta}_{i,\varepsilon}$ is the corrupted sub-optimality gap, which is smaller than the uncorrupted gap $\Delta$ and thus, harder to distinguish. To complement the lower bounds, we design a robust algorithm `HuberUCB` that uses Huber's estimator for robust mean estimation and a novel concentration bound on this estimator to create tight confidence intervals. `HuberUCB` achieves logarithmic regret that matches the lower bound for low suboptimality gap/high variance regime. We also present a sequential Huber estimator that could be of independent interest and we use it to state a linear-time robust bandit algorithm, `SeqHuberUCB`, that presents the same efficiency as `HuberUCB`. Unlike existing literature, we do not need any assumption on a known bound on corruption and a known bound on the $(1+\eta)$-uncentered moment, which was posed as an open problem in Agrawal et al. (2021).

Since our upper and lower bounds disagree in the high gap/low variance regime, it will be interesting to investigate this regime further. From multi-armed bandits, we know that the tightest lower and upper bounds depend on the KL-divergence between optimal and suboptimal reward distributions. Thus, it would be imperative to study KL-divergence with corrupted distributions to better understand the Bandits with Stochastic Corruption problem. In this paper, we have focused on a problem-dependent regret analysis for a given $\varepsilon$. In future, it would be interesting to get some insight on how to adapt to an unknown $\varepsilon$, and to perform a problem-independent "worst-case" analysis. Also, following the reinforcement learning literature, it will be natural to extend `HuberUCB` to contextual and linear bandit settings with corruptions and heavy-tails. This will facilitate its applicability to practical problems, such as choosing treatments against pests.

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

# Appendix

## Table of Contents

# A   Proof of Theorems

## A.1   Proof of Theorem 1: Regret Lower Bound

The theorem is a consequence of Lemmas 2, 3 and 4.
From Lemma 2, we have

$$\lim_{n\to\infty} \inf \frac{\mathbb{E}_\nu[T_i(n)]}{\log(n)} \geq \frac{1}{D_{\mathrm{KL}}(P_0, P_1)} \tag{16}$$

**Student distributions**

Let $P_0, P_1$ be student distributions with parameter $d = 3$ and gap $\Delta_i$ as in Lemma 3. From Lemma 3, we get

$$D_{\mathrm{KL}}(P_0, P_1) \leq \begin{cases} 17\Delta_i^2 & \text{if } \Delta_i \leq 1 \\ 4\log(\Delta_i) + \log(50) & \text{if } \Delta_i > 1 \end{cases} \tag{17}$$

Then, using that $\log(50) \leq 17$,

$$D_{\mathrm{KL}}(P_0, P_1) \leq 17\Delta_i^2 \wedge 4\log(\Delta_i) + 17.$$

Finally, use that the variance of a student with three degrees of freedom is $\sigma_i^2 = 3$ to get that

$$D_{\mathrm{KL}}(P_0, P_1) \leq 51\frac{\Delta_i^2}{\sigma_i^2} \wedge 4\log\left(\frac{\Delta_i}{\sigma_i}\right) + 22.$$

**Bernoulli distributions**

Let $P_0, P_1$ be as in Lemma 4 with gap $\Delta_i$ and variance $\sigma_i$. If $2\sigma_i \frac{\varepsilon}{\sqrt{1-2\varepsilon}} < \Delta_i < 2\sigma_i$, then

$$D_{\mathrm{KL}}(P_0, P_1) \leq \frac{\overline{\Delta}_{i,\varepsilon}}{2\sigma_i}\log\left(1 + \frac{\overline{\Delta}_{i,\varepsilon}}{2\sigma_i - \overline{\Delta}_{i,\varepsilon}}\right) \wedge (1 - 2\varepsilon)\log\left(1 + \frac{1 - 2\varepsilon}{\varepsilon}\right) \tag{18}$$

Use Equation (16) to conclude.

## A.2   Proof of Theorem 2: Concentration of Huber's Estimator

First, we control the deviations of Huber's estimator using the deviations of $\psi_\beta(X - \mathrm{Hub}_\beta(X_1^n))$. We will need the following lemma to control the variance of $\psi_\beta(X - \mathrm{Hub}_\beta(X_1^n))$, which will in turn allow us to control its deviation with Lemma 8.

**Lemma 7 (Controlling Variance of Influence of Huber's Estimator)** *Suppose that $Y_1, \ldots, Y_n$ are i.i.d with law $P$. Then*

$$\mathrm{Var}(\psi_\beta(Y - \mathrm{Hub}_\beta(P))) \leq \mathrm{Var}(Y) = \sigma^2$$

**Lemma 8 (Concentrating Huber's Estimator by Concentrating the Influence)** *Suppose that $X_1$, $\ldots, X_n$ are i.i.d with law $(1 - \varepsilon)P + \varepsilon H$ for some $H \in \mathcal{P}$ and proportion of outliers $\varepsilon \in (0, 1/2)$. Then, for any $\eta > 0$ and $\lambda \in (0, \beta/2]$, we have*

$$\mathbb{P}(|\mathrm{Hub}_\beta(X_1^n) - \mathrm{Hub}_\beta(P)| \geq \lambda) \leq \mathbb{P}\left(\left|\frac{1}{n}\sum_{i=1}^n \psi_\beta(X_i - \mathrm{Hub}_\beta(P))\right| \geq \lambda(p - \eta - \varepsilon)_+\right) + 2e^{-2n\eta^2}$$

*where $p = \mathbb{P}(|Y - \mathbb{E}[X]| \leq \beta/2)$.*

Then, using these Lemmas, we can prove the theorem.

**Step 1.** For any $\delta \in (0,1)$, with probability larger than $1 - 3\delta$,

$$\left| \frac{1}{n} \sum_{i=1}^{n} \psi_\beta(X_i - \mathrm{Hub}_\beta(P)) \right| \le \sigma \sqrt{\frac{2 \log(1/\delta)}{n}} + \beta \frac{\log(1/\delta)}{2n} + 2\beta\varepsilon + 2\beta \sqrt{\frac{\log(1/\delta)(1 - 2\varepsilon)}{n \log\left(\frac{1-\varepsilon}{\varepsilon}\right)}}. \qquad (19)$$

PROOF: Write that $X_i = (1 - W_i)Y_i + W_i Z_i$ where $W_1, \ldots, W_n$ are i.i.d $\{0,1\}$ Bernoulli random variable with mean $\varepsilon$, $Y_1, \ldots, Y_n$ are i.i.d $\sim P$ and $Z_1, \ldots, Z_n$ are i.i.d with law $H$, we have

$$\left| \frac{1}{n} \sum_{i=1}^{n} \psi_\beta(X_i - \mathrm{Hub}_\beta(P)) \right|$$

$$= \left| \frac{1}{n} \sum_{i=1}^{n} \psi_\beta(Y_i - \mathrm{Hub}_\beta(P)) + \frac{1}{n} \sum_{i=1}^{n} \mathbf{1}\{W_i = 1\} \left( \psi_\beta(Z_i - \mathrm{Hub}_\beta(P)) - \psi_\beta(Y_i - \mathrm{Hub}_\beta(P)) \right) \right|$$

$$\le \left| \frac{1}{n} \sum_{i=1}^{n} \psi_\beta(Y_i - \mathrm{Hub}_\beta(P)) \right| + 2\beta \frac{1}{n} \sum_{i=1}^{n} \mathbf{1}\{W_i = 1\}$$

Remark that by definition of $\mathrm{Hub}_\beta(P)$, it is defined as the root of the equation $\mathbb{E}[\psi_\beta(Y - \mathrm{Hub}_\beta(P))] = 0$. From Bernstein's inequality, for any $\delta \in (0,1)$,

$$\mathbb{P}\left( \left| \frac{1}{n} \sum_{i=1}^{n} \psi_\beta(Y_i - \mathrm{Hub}_\beta(P)) \right| \ge \sqrt{\frac{2 V_{\psi_\beta} \log(1/\delta)}{n}} + \beta \frac{\log(1/\delta)}{3n} \right) \le 2\delta$$

where $V_{\psi_\beta} = \mathrm{Var}(\psi_\beta(Y_i - \mathrm{Hub}_\beta(P)))$.

Then, using that Bernoulli random variables with mean $\varepsilon$ are sub-Gaussian with variance parameter $\frac{1-2\varepsilon}{2 \log((1-\varepsilon)/\varepsilon)}$ (see Lemma 6 of Bourel et al. (2020)),

$$\mathbb{P}\left( \frac{1}{n} \sum_{i=1}^{n} \mathbf{1}\{W_i = 1\} \le \varepsilon + \sqrt{\frac{\log(1/\delta)(1 - 2\varepsilon)}{n \log\left(\frac{1-\varepsilon}{\varepsilon}\right)}} \right) \ge 1 - \delta.$$

Then, using Lemma 7 we get for any $\delta \in (0,1)$, with probability larger than $1 - 3\delta$,

$$\left| \frac{1}{n} \sum_{i=1}^{n} \psi_\beta(X_i - \mathrm{Hub}_\beta(P)) \right| \le \sigma \sqrt{\frac{2 \log(1/\delta)}{n}} + \beta \frac{\log(1/\delta)}{2n} + 2\beta\varepsilon + 2\beta \sqrt{\frac{\log(1/\delta)(1 - 2\varepsilon)}{n \log\left(\frac{1-\varepsilon}{\varepsilon}\right)}}. \qquad (20)$$

**Step 2.** Using $\eta = \sqrt{\frac{\log(1/\delta)}{2n}}$, the hypotheses of Lemma 8 are verified.

PROOF: To apply Lemma 8, it is sufficient that

$$\sigma \sqrt{\frac{2t}{n}} + \beta \frac{\log(1/\delta)}{3n} + 2\beta\varepsilon + 2\beta \sqrt{\frac{\log(1/\delta)(1 - 2\varepsilon)}{n \log\left(\frac{1-\varepsilon}{\varepsilon}\right)}} \le \frac{\beta}{2} \left( p - \sqrt{\frac{\log(1/\delta)}{2n}} - \varepsilon \right) \qquad (21)$$

and using that $4\sigma \le \beta$, we have that it is sufficient that

$$\sqrt{\frac{\log(1/\delta)}{2n}} + \frac{\log(1/\delta)}{3n} + 2 \sqrt{\frac{\log(1/\delta)(1 - 2\varepsilon)}{n \log\left(\frac{1-\varepsilon}{\varepsilon}\right)}} \le \frac{1}{2} \left( p - 5\varepsilon \right). \qquad (22)$$

This is a polynomial in $\sqrt{\log(1/\delta)/n}$ that we need to solve. We use the following elementary algebra lemma.

**Lemma 9 (2nd order polynomial root bound)** *let $a, b, c$ be three positive constants and $x$ verify $ax^2 + bx - c \le 0$. Suppose that $\frac{4ac}{b^2} \le d$, then $x$ must verify*

$$x \ge \frac{2c(\sqrt{d+1} - 1)}{db}.$$

Observe that we have

$$\frac{2\left(p - 5\varepsilon\right)}{3\left(\frac{1}{\sqrt{2}} + \frac{2\sqrt{1-2\varepsilon}}{\sqrt{\log\left(\frac{1-\varepsilon}{\varepsilon}\right)}}\right)^2} \leq \frac{4}{3}$$

and $(\sqrt{4/3 + 1} - 1)/(4/3) \geq 8/7$, hence, from Lemma 9, we get the following sufficient condition for Equation (22) to hold:

$$\sqrt{\log(1/\delta)/n} \leq \frac{8\sqrt{2}\left(p - 5\varepsilon\right)}{7\left(1 + \frac{2\sqrt{2(1-2\varepsilon)}}{\sqrt{\log\left(\frac{1-\varepsilon}{\varepsilon}\right)}}\right)}.$$

Hence, taking this to the square,

$$\log(1/\delta) \leq n\frac{128\left(p - 5\varepsilon\right)^2}{49\left(1 + \frac{2\sqrt{2(1-2\varepsilon)}}{\sqrt{\log\left(\frac{1-\varepsilon}{\varepsilon}\right)}}\right)^2}.$$

**Step 3.** Using Lemma 8 and Step 1 prove that the theorem is true.

PROOF: The hypotheses of Lemma 8 are verified and we can use its result and together with Equation (19) we get with probability larger than $1 - 5\delta$,

$$|\text{Hub}_\beta(X_1^n) - \text{Hub}_\beta(P)| \leq \frac{\sigma\sqrt{\frac{2\log(1/\delta)}{n}} + \beta\frac{\log(1/\delta)}{3n} + 2\beta\sqrt{\frac{\log(1/\delta)(1-2\varepsilon)}{n\log\left(\frac{1-\varepsilon}{\varepsilon}\right)}} + 2\beta\varepsilon}{\left(p - \sqrt{\frac{\log(1/\delta)}{2n}} - \varepsilon\right)_+}.$$

### A.3 Proof of Theorem 4: Concentration of Sequential Huber's Estimator

In this proof, we denote

$$r_t(\delta) := \frac{\sigma\sqrt{\frac{2\log(1/\delta)}{t}} + \beta\frac{\log(1/\delta)}{3t} + 2\beta\overline{\varepsilon}\sqrt{\frac{\log(1/\delta)}{t}} + 2\beta\varepsilon}{\left(p - \sqrt{\frac{\log(1/\delta)}{2t}} - \varepsilon\right)_+}$$

this is the rate of convergence of $\text{Hub}_\beta(X_1^t)$ to $\text{Hub}_\beta(P)$, as stated by Theorem 2.

Let $P_2(t) < t < P_2(t+1)$, define

$$f_t(u) = \frac{1}{t}\sum_{i=1}^{t}\psi_\beta(X_i - u).$$

$f_t$ is a continuous function, we take its derivative in distribution to get that

$$f_t(\text{Hub}_\beta(P)) = f_t(H_t) + (\text{Hub}_\beta(P) - H_t)f_t'(H_t) + \int_{H_t}^{\text{Hub}_\beta(P)} f_t''(u)\left(\text{Hub}_\beta(P) - u\right)\mathrm{d}u$$

Then, by definition of $\text{SeqHub}_t$, we also have

$$0 = f_t(H_t) + (\text{SeqHub}_t - H_t)f_t'(H_t).$$

Hence,

$$f_t(Hub(P)) = (\text{Hub}_\beta(P) - \text{SeqHub}_t)f_t'(H_t) + \int_{H_t}^{\text{Hub}_\beta(P)} f_t''(u)\left(\text{Hub}_\beta(P) - u\right)\mathrm{d}u. \tag{23}$$

where $f_t'(u) = -\frac{1}{t}\sum_{i=1}^{t}\mathbf{1}\{|X_i - u| \leq \beta\}$ and $f_t''(u) = -\frac{1}{t}\sum_{i=1}^{t}(\delta_{X_i-u-\beta} - \delta_{X_i-u+\beta})$ where $\delta_x$ is the Dirac mass in $x$.

$f'_t(H_t)$ is a sum of indicator functions and should be close to $\mathbb{P}(|X - \mathbb{E}[X]| \leq \beta)$, which is close to 1.

**Bound on $f'_t(H_t)$**

We bound $|f'_t(H_t)|$. We have

$$|f'_t(H_t)| = \frac{1}{t} \sum_{i=1}^{t} \mathbf{1}\{|X_i - H_t| \leq \beta\}$$

$$\geq \frac{1}{t} \sum_{i=1}^{t} \mathbf{1}\{|X_i - \mathrm{Hub}_\beta(P)| \leq \beta - |H_t - \mathrm{Hub}_\beta(P)|\}.$$

Choose the limiting $\delta$ which is $\delta = \exp\left(-P_2(t)\frac{128(p-5\varepsilon)^2}{49\left(1+2\overline{\varepsilon}\sqrt{2}\right)^2}\right)$, from Equation (21), we get that $r_t(\delta) \leq \beta/2$.

Then, we have from Theorem 2, with probability larger than $1 - 5\exp\left(-P_2(t)\frac{128(p-5\varepsilon)^2}{49\left(1+2\overline{\varepsilon}\sqrt{2}\right)^2}\right)$, that $|H_t - \mathrm{Hub}_\beta(P)| \leq \beta/2$, and then,

$$|f'_t(H_t)| \geq \frac{1}{t} \sum_{i=1}^{t} \mathbf{1}\{|X_i - \mathrm{Hub}_\beta(P)| \leq \beta/2\}. \tag{24}$$

**Bound on the integral of $f''_t$.**

We have,

$$\int_{H_t}^{\mathrm{Hub}_\beta(P)} f''_t(u) \left(\mathrm{Hub}_\beta(P) - u\right)\mathrm{d}u$$

$$= \frac{1}{t} \sum_{i=1}^{t} \int_{H_t}^{\mathrm{Hub}_\beta(P)} \left(\delta_{X_i - u - \beta} - \delta_{X_i - u + \beta}\right)\left(\mathrm{Hub}_\beta(P) - u\right)\mathrm{d}u$$

$$= \frac{1}{t} \sum_{i=1}^{t} (\mathrm{Hub}_\beta(P) - X_i - \beta)\mathbf{1}\{X_i \in I_-\} - (\mathrm{Hub}_\beta(P) - X_i + \beta)\mathbf{1}\{X_i \in I_+\}$$

where $I_-$ and $I_+$ are the two undirected intervals

$$I_- = [H_t - \beta, \mathrm{Hub}_\beta(P) - \beta] \quad \text{and} \quad I_+ = [H_t + \beta, \mathrm{Hub}_\beta(P) + \beta].$$

Figure 3: Illustration $I_-$ and $I_+$

Having that $|\mathrm{Hub}_\beta(X_1^t) - H_t| \leq \beta/2$, we have that $I_- \cap I_+ = \emptyset$. Then, choosing either the sum $\frac{1}{t}\sum_{i=1}^{t}(\mathrm{Hub}_\beta(P) - X_i - \beta)\mathbf{1}\{X_i \in I_-\}$ or $\frac{1}{t}\sum_{i=1}^{t}(\mathrm{Hub}_\beta(P) - X_i - \beta)\mathbf{1}\{X_i \in I_+\}$ according to which one is larger. If $X_i \in I_+$, we have $|\mathrm{Hub}_\beta(P) - X_i + \beta| \leq |\mathrm{Hub}_\beta(P) - H_t|$ and if $X_i \in I_-$, $|\mathrm{Hub}_\beta(P) - X_i - \beta| \leq |\mathrm{Hub}_\beta(P) - H_t|$, hence we have

$$\left|\frac{1}{t}\sum_{i=1}^{t}\int_{H_t}^{\mathrm{Hub}_\beta(P)}\left(\delta_{X_i - u - \beta} - \delta_{X_i - u + \beta}\right)\left(\mathrm{Hub}_\beta(P) - u\right)\mathrm{d}u\right|$$

$$\leq |\mathrm{Hub}_\beta(P) - H_t|\max\left(\frac{1}{t}\sum_{i=1}^{t}\mathbf{1}\{X_i \in I_-\}, \frac{1}{t}\sum_{i=1}^{t}\mathbf{1}\{X_i \in I_+\}\right).$$

Now, remark that by Equation (24), we have,

$$\sum_{i=1}^{t} \mathbf{1}\{X_i - H_t\} = |f'_t(H_t)| \geq \frac{1}{t}\sum_{i=1}^{t}\mathbf{1}\{|X_i - \text{Hub}_\beta(P)| \leq \beta/2\}$$

Let us denote $p_t(\beta) = \frac{1}{t}\sum_{i=1}^{t}\mathbf{1}\{|X_i - \text{Hub}_\beta(P)| \leq \beta/2\}$.

There cannot be more than $1 - p_t(\beta)$ fraction of the $X_i$'s that are outside $[H_t - \beta, H_t + \beta]$. Similarly, there cannot be more than $1 - p_t(\beta)$ fraction of the $X'_i s$ that are outside $[\text{Hub}_\beta(P) - \beta/2, \text{Hub}_\beta(P) + \beta/2]$. Hence, if $H_t \leq \text{Hub}_\beta(P)$, then $I_- \subset [H_t - \beta, H_t + \beta]^c$ and the proportion of $X_i$'s in $I_-$ can't be larger than $1 - p_t(\beta)$.

If $\text{Hub}_\beta(P) \leq H_t$, then $I_- \subset [\text{Hub}_\beta(P) - \beta, \text{Hub}_\beta(P) + \beta]^c$ which is itself a subset of $[\text{Hub}_\beta(P) - \beta/2, \text{Hub}_\beta(P) + \beta/2]^c$ and the proportion of $X_i$'s included in $[\text{Hub}_\beta(P) - \beta/2, \text{Hub}_\beta(P) + \beta/2]^c$ cannot be larger than $3/10$.

In both cases, $\frac{1}{t}\sum_{i=1}^{t}\mathbf{1}\{X_i \in I_-\} \leq 1 - p_t(\beta)$. A similar reasoning holds for $I_+$, hence

$$\left|\int_{H_t}^{\text{Hub}_\beta(P)} f''_t(u)\left(\text{Hub}_\beta(P) - u\right)\mathrm{d}u\right| \leq (1 - p_t(\beta))|\text{Hub}_\beta(P) - H_t|$$

Then, using Equation (24) and Equation (23), we get with probability larger than $1 - 5\exp\left(-P_2(t)\frac{128(p-5\varepsilon)^2}{49\left(1+2\overline{\varepsilon}\sqrt{2}\right)^2}\right)$,

$$|\text{Hub}_\beta(P) - \text{SeqHub}_t| \leq \frac{f_t(\text{Hub}_\beta(P)) + \left|\int_{H_t}^{\text{Hub}_\beta(P)} f''_t(u)\left(\text{Hub}_\beta(P) - u\right)\mathrm{d}u\right|}{f'_t(H_t)}$$
$$\leq \frac{f_t(\text{Hub}_\beta(P)) + (1 - p_t(\beta))|\text{Hub}_\beta(P) - H_t|}{p_t(\beta)}. \tag{25}$$

Then let $\delta \leq \exp\left(-P_2(t)\frac{128(p-5\varepsilon)^2}{49\left(1+2\overline{\varepsilon}\sqrt{2}\right)^2}\right)$, we use Equation (20) to say that with probability larger than $1 - 3\delta$, we have

$$f_t(\text{Hub}_\beta(P)) \leq \sigma\sqrt{\frac{2\log(1/\delta)}{t}} + \beta\frac{\log(1/\delta)}{2t} + 2\beta\varepsilon + 2\beta\sqrt{\frac{\log(1/\delta)(1-2\varepsilon)}{t\log\left(\frac{1-\varepsilon}{\varepsilon}\right)}}.$$

Then, using Hoeffding's inequality after taking out the outliers, we get with probability larger than $1 - \delta$, that

$$p_t(\beta) = \frac{1}{t}\sum_{i=1}^{t}\mathbf{1}\{|X_i - \text{Hub}(P)| \leq \beta/2\} \geq p - \sqrt{\frac{\log(1/\delta)}{2t}} - \varepsilon$$

to recover that the first term of the right-hand-side of Equation (25) is smaller than $r_t(\delta)$. Then, using Theorem 2, we get that with probability larger than $1 - 5\exp\left(-P_2(t)\frac{128(p-5\varepsilon)^2}{49\left(1+2\overline{\varepsilon}\sqrt{2}\right)^2}\right) - 9\delta \geq 1 - 14\delta$,

$$|\text{Hub}_\beta(P) - \text{SeqHub}_t| \leq r_t(\delta) + \left(\frac{1}{p - \sqrt{\frac{\log(1/\delta)}{2t}} - \varepsilon} - 1\right)r_{P_2(t)}(\delta).$$

### A.4 Proof of Theorem 3: Regret Upper bound of HuberUCB

If $A_t = i$ then at least one of the following four inequalities is true:

$$\widehat{\text{Hub}}_{1,T_1(t-1)} + B_1(T_1(t-1), t) \leq \mu_1 \tag{26}$$

or

$$\widehat{\text{Hub}}_{i,T_i(t-1)} \geq \mu_i + B_i(T_i(t-1), t) \tag{27}$$

or

$$\Delta_i < 2B_i(T_i(t-1), t) \tag{28}$$

or

$$T_1(t-1) < s_{lim}(t) = \frac{98\log(t)}{128\,(p-5\varepsilon)^2}\left(1 + 2\sqrt{2}\left(\overline{\varepsilon} \vee \frac{9}{14\sqrt{2}}\right)\right)^2 \tag{29}$$

Indeed, if $T_i(t-1) < s_{lim}(t)$, then $B_i(T_i(t-1), t) = \infty$ and Inequality (28) is true. On the other hand, if $T_i(t-1) \geq s_{lim}(t)$, then we have $B_i(T_i(t-1), t)$ is finite and all four inequalities are false, then,

$$\begin{aligned}
\widehat{\mathrm{Hub}}_{1,T_1(t-1)} + B_1(T_1(t-1), t) &> \mu_1 \\
&= \mu_i + \Delta_i \\
&\geq \mu_i + 2B_i(T_i(t-1), n) \\
&\geq \mu_i + 2B_i(T_i(t-1), t) \\
&\geq \widehat{\mathrm{Hub}}_{i,T_i(t-1)} + B_i(T_i(t-1), t)
\end{aligned}$$

which implies that $A_t \neq i$.

**Step 1.** We have that $\mathbb{P}\left((26) \text{ is true}\right) \leq 5/t$.
PROOF:

Then, we have that,

$$\begin{aligned}
\mathbb{P}\left(\widehat{\mathrm{Hub}}_{1,T_1(t-1)} + B_1(T_1(t-1), t) \leq \mu_1\right) &\leq \sum_{s=1}^{t} \mathbb{P}\left(\widehat{\mathrm{Hub}}_{1,s} + B_1(s, t) \leq \mu_1\right) \\
&= \sum_{s=\lceil s_{lim}(t)\rceil}^{t} \mathbb{P}\left(\widehat{\mathrm{Hub}}_{1,s} - \mu_1 \leq -B_1(s, t)\right)
\end{aligned}$$

Then, use Theorem 2, we get

$$\begin{aligned}
\mathbb{P}\left(\widehat{\mathrm{Hub}}_{1,T_1(t-1)} + B_1(T_1(t-1), t) \leq \mu_1\right) &\leq \sum_{s=\lceil s_{lim}(t)\rceil}^{t} 5e^{-\log(t^2)} \\
&\leq \sum_{s=\lceil s_{lim}(t)\rceil}^{t} \frac{5}{t^2} \leq \frac{5}{t}.
\end{aligned}$$

**Step 2.** Similarly, for arm $i$, we have

$$\mathbb{P}\left(\widehat{\mathrm{Hub}}_{i,T_i(t-1)} \geq \mu_i + B_i(T_i(t-1), t)\right) \leq \frac{5}{t}$$

PROOF: We have,

$$\begin{aligned}
\mathbb{P}\left(\widehat{\mathrm{Hub}}_{i,T_i(t-1)} \geq \mu_i + B_i(T_i(t-1), t)\right) &\leq \sum_{s=\lceil s_{lim}(t)\rceil}^{t} \mathbb{P}\left(\widehat{\mathrm{Hub}}_{i,s} - \mu_i \geq B_i(s, t)\right) \\
&\leq \sum_{s=\lceil s_{lim}(t)\rceil}^{t} 5e^{-\log(t^2)} \leq \frac{5}{t}.
\end{aligned}$$

**Step 3.** Let $v \in \mathbb{N}$. If one of the two following conditions are true, then for all $t$ such that $T_i(t-1) \geq v$, we have $\Delta_i \geq 2B_i(T_i(t-1), t)$ (i.e. Equation (28) is false).

Condition 1: if $\widetilde{\Delta}_{i,\varepsilon} > 12 \frac{\sigma_i^2}{\beta_i} \left( \sqrt{2} + 2 \frac{\beta_i}{\sigma_i} \overline{\varepsilon} \right)^2$ and $v \leq \log(n) \frac{96\beta_i}{9\widetilde{\Delta}_{i,\varepsilon}}$.

Condition 2: if $\widetilde{\Delta}_{i,\varepsilon} \leq 12 \frac{\sigma_i^2}{\beta_i} \left( \sqrt{2} + 2 \frac{\beta_i}{\sigma_i} \overline{\varepsilon} \right)^2$ and $v \leq \frac{50}{9\widetilde{\Delta}_{i,\varepsilon}^2} \left( \sigma_i \sqrt{2} + 2\beta_i \overline{\varepsilon} \right)^2 \log(n)$.

PROOF: We search for the smallest value $v \geq s_{lim}(t)$ such that $\Delta_i$ verifies

$$\Delta_i \geq 2B_i(v, t) = 2 \frac{\sigma_i \sqrt{\frac{2\log(t^2)}{v}} + \beta \frac{\log(t^2)}{3v} + 2\overline{\varepsilon}\beta_i \sqrt{\frac{\log(t^2)}{v}} + 2\beta_i\varepsilon}{\left( p - \sqrt{\frac{\log(t^2)}{2v}} - \varepsilon \right)} + 2b_i.$$

First, we simplify the expression, having that $v \geq s_{lim}(t)$, we have

$$\frac{\log(t^2)}{2v} \leq \frac{128(p - 5\varepsilon)^2}{98(1 + 9/7)^2} \leq \frac{(p - \varepsilon)^2}{4},$$

hence we simplify to

$$\Delta_i \geq \frac{4}{(p - \varepsilon)} \left( \sigma_i \sqrt{\frac{2\log(t^2)}{v}} + \beta_i \frac{\log(t^2)}{3v} + 2\beta_i\overline{\varepsilon}\sqrt{\frac{\log(t^2)}{v}} + 2\beta_i\varepsilon \right) + 2b_i$$

let us denote $\widetilde{\Delta}_{i,\varepsilon} = (\Delta_i - 2b_i)(p - \varepsilon) - 8\beta_i\varepsilon$, we are searching for $v$ such that

$$\beta_i \frac{\log(t^2)}{3v} + \sqrt{\frac{\log(t^2)}{v}} \left( \sigma_i \sqrt{2} + 2\beta_i\overline{\varepsilon} \right) - \frac{\widetilde{\Delta}_{i,\varepsilon}}{4} \leq 0$$

This is a second order polynomial in $\sqrt{\log(t^2)/v}$.

If $\widetilde{\Delta}_{i,\varepsilon} > 0$, then the smallest $v > 0$ is

$$\sqrt{\frac{\log(t^2)}{v}} = \frac{3}{2\beta_i} \left( -\left( \sigma_i\sqrt{2} + 2\overline{\varepsilon}\beta_i \right) + \sqrt{\left( \sigma_i\sqrt{2} + 2\beta_i\overline{\varepsilon} \right)^2 + \frac{\widetilde{\Delta}_{i,\varepsilon}\beta_i}{3}} \right).$$

**First setting:** if $\widetilde{\Delta}_{i,\varepsilon} > 12 \frac{\sigma_i^2}{\beta_i} \left( \sqrt{2} + 2 \frac{\beta_i}{\sigma_i} \overline{\varepsilon} \right)^2$,

In that case, we have

$$\sqrt{\frac{\log(t^2)}{v}} \geq \frac{3}{2\beta_i} \left( -\left( \sigma_i\sqrt{2} + 2\beta_i\overline{\varepsilon} \right) + \sqrt{\frac{\beta_i\widetilde{\Delta}_{i,\varepsilon}}{3}} \right) \geq \frac{3}{2\beta_i} \sqrt{\frac{\beta_i\widetilde{\Delta}_{i,\varepsilon}}{12}} = \sqrt{\frac{9\widetilde{\Delta}_{i,\varepsilon}}{48\beta_i}}$$

Hence, $v \leq \log(t) \frac{96\beta_i}{9\widetilde{\Delta}_{i,\varepsilon}}$.

**Second setting:** if $\widetilde{\Delta}_{i,\varepsilon} \leq 12 \frac{\sigma_i^2}{\beta_i} \left( \sqrt{2} + 2 \frac{\beta_i}{\sigma_i} \overline{\varepsilon} \right)^2$, then we use Lemma 9, using that

$$\frac{\widetilde{\Delta}_{i,\varepsilon}\beta_i}{3 \left( \sigma_i\sqrt{2} + 2\beta_i\overline{\varepsilon} \right)^2} \leq 4$$

and the fact that $\frac{\sqrt{1+4}-1}{4} \geq \frac{3}{10}$, we get,

$$\sqrt{\frac{\log(t^2)}{v}} \geq \frac{3\widetilde{\Delta}_{i,\varepsilon}}{5 \left( \sigma_i\sqrt{2} + 2\beta_i\overline{\varepsilon} \right)}$$

Hence,

$$v \leq \frac{50}{9\widetilde{\Delta}_{i,\varepsilon}^2} \left( \sigma_i \sqrt{2} + 2\beta_i \bar{\varepsilon} \right)^2 \log(t).$$

**Step 4.** Using All the previous steps, we prove the theorem. PROOF: We have

$$
\begin{aligned}
\mathbb{E}[T_i(t)] &= \mathbb{E}\left[ \sum_{t=1}^{t} \mathbf{1}\{A_t = i\} \right] \\
&\leq \lfloor \max(v, s_{lim}(t)) \rfloor + \mathbb{E}\left[ \sum_{t=\lfloor \max(v, s_{lim}(t)) \rfloor + 1}^{t} \mathbf{1}\{A_t = i \text{ and (28) is false}\} \right] \\
&\leq \lfloor \max(v, s_{lim}(t)) \rfloor + \mathbb{E}\left[ \sum_{t=\lfloor \max(v, s_{lim}(t)) \rfloor + 1}^{t} \mathbf{1}\{(26) \text{ or } (27) \text{ or } (29) \text{ is true}\} \right] \\
&= \lfloor \max(v, s_{lim}(t)) \rfloor + \sum_{t=\lfloor \min(v, s_{lim}(t)) \rfloor + 1}^{t} \mathbb{P}\left( (26) \text{ or } (27) \text{ is true} \right) \\
&\leq \lfloor \max(v, s_{lim}(t)) \rfloor + 2 \sum_{t=\lfloor \min(v, s_{lim}(t)) \rfloor + 1}^{t} \frac{5}{t}
\end{aligned}
$$

using the harmonic series bound by $\log(t) + 1$, we have

$$\mathbb{E}[T_i(t)] \leq \max(v, s_{lim}(t)) + 10(\log(t) + 1)$$

Then, we replace the value of $v$,

**First setting:** $\widetilde{\Delta}_{i,\varepsilon} > 12 \frac{\sigma_i^2}{\beta_i} \left( \sqrt{2} + 2\frac{\beta_i}{\sigma_i} \bar{\varepsilon} \right)^2$

$$\mathbb{E}[T_i(t)] \leq \log(t) \max \left( \frac{96\beta_i}{9\widetilde{\Delta}_{i,\varepsilon}}, \frac{4}{(p-5\varepsilon)^2} \left( 1 + 2\sqrt{2} \left( \bar{\varepsilon} \vee \frac{9}{14\sqrt{2}} \right) \right)^2 \right) + 10(\log(t) + 1)$$

**Second setting:** if $\widetilde{\Delta}_{i,\varepsilon} \leq 12 \frac{\sigma_i^2}{\beta_i} \left( \sqrt{2} + 2\frac{\beta_i}{\sigma_i} \bar{\varepsilon} \right)^2$, then

$$\mathbb{E}[T_i(t)] \leq \log(n) \max \left( \frac{50}{9\widetilde{\Delta}_{i,\varepsilon}^2} \left( \sigma_i \sqrt{2} + 2\beta_i \bar{\varepsilon} \right)^2, \frac{4}{(p-5\varepsilon)^2} \left( 1 + 2\sqrt{2} \left( \bar{\varepsilon} \vee \frac{9}{14\sqrt{2}} \right) \right)^2 \right) + 10(\log(t) + 1).$$

This concludes the proof of Theorem 3.

# B   Proof of Technical Lemmas and Corollaries

## B.1   Preliminary lemmas

### B.1.1   Proof of Lemma 1: Regret Decomposition

From Equation (Corrupted regret), we have

$$R_n = \sum_{a=1}^{k} \sum_{t=1}^{n} \mathbb{E}\left[ (\max_a \mathbb{E}_{P_a}[X'] - X_t')\mathbf{1}\{A_t = a\} \right]$$

Then, we condition on $A_t$

$$
\begin{aligned}
\mathbb{E}\left[ (\max_a \mathbb{E}_{P_a}[X'] - X_t')\mathbf{1}\{A_t = a\} \,|A_t \right] &= \mathbf{1}\{A_t = a\}\mathbb{E}[\max_a \mathbb{E}_{P_a}[X'] - X_t'|A_t] \\
&= \mathbf{1}\{A_t = a\}(\max_a \mathbb{E}_{P_a}[X'] - \mu_{A_t}) \\
&= \mathbf{1}\{A_t = a\}(\max_a \mathbb{E}_{P_a}[X'] - \mu_a) = \mathbf{1}\{A_t = a\}\Delta_a
\end{aligned}
$$

and this stays true whatever the policy, because the policy at time $t$ use knowledge up to time $t-1$, hence its decision does not depend on $X_t$. Hence, we have

$$R_n(\pi) = \sum_{a=1}^{k} \Delta_a \mathbb{E}_{\pi(\cdot|X_1^n, A_1^n)}[T_a(n)]$$

where $T_a(n)$ is with respect to the randomness of $\pi$, which is to say that we compute $\mathbb{E}[T_i(n)]$ in the corrupted setting and not in the uncorrupted one.

$$R_n = \sum_{a=1}^{k} \Delta_a \mathbb{E}_{\nu_\varepsilon}[T_a(n)].$$

### B.1.2   Proof of Lemma 3: KL for Student's Distribution

First, we compute the $\chi^2$ divergence between the two laws $f_a$ and $f_0$. We have, for any $a \geq 0$

$$
d_{\chi^2}(f_a, f_0) = \int \frac{(f_a(x) - f_0(x))^2}{f_0(x)} dx
$$

$$
= \frac{\Gamma\left(\frac{d+1}{2}\right)}{\Gamma\left(\frac{d}{2}\right)\sqrt{d\pi}} \int_{\mathbb{R}} \left( \frac{1}{\left(1 + \frac{(x-a)^2}{d}\right)^{\frac{d+1}{2}}} - \frac{1}{\left(1 + \frac{x^2}{d}\right)^{\frac{+1}{2}}} \right)^2 \left(1 + \frac{x^2}{d}\right)^{\frac{d+1}{2}} dx
$$

$$
= \frac{\Gamma\left(\frac{d+1}{2}\right)}{\Gamma\left(\frac{d}{2}\right)\sqrt{d\pi}} \int_{\mathbb{R}} \frac{\left( \left(1 + \frac{(x-a)^2}{d}\right)^{\frac{d+1}{2}} - \left(1 + \frac{x^2}{d}\right)^{\frac{d+1}{2}} \right)^2}{\left(1 + \frac{(x-a)^2}{d}\right)^{d+1} \left(1 + \frac{x^2}{d}\right)^{\frac{d+1}{2}}} dx
$$

$$
= \frac{\Gamma\left(\frac{d+1}{2}\right)}{\Gamma\left(\frac{d}{2}\right)\sqrt{d\pi}} \left( \int_{\mathbb{R}} \frac{dx}{\left(1 + \frac{x^2}{d}\right)^{\frac{d+1}{2}}} - 2\int_{\mathbb{R}} \frac{dx}{\left(1 + \frac{(x-a)^2}{d}\right)^{\frac{d+1}{2}}} + \int_{\mathbb{R}} \frac{\left(1 + \frac{x^2}{d}\right)^{\frac{d+1}{2}}}{\left(1 + \frac{(x-a)^2}{d}\right)^{d+1}} dx \right).
$$

The first two terms are respectively equal to 1 and $-2$ using the fact that the student distribution integrate to 1. Then, we do the change of variable $y = x - a$ in the last integral to get

$$
d_{\chi^2}(f_a, f_0) = \frac{\Gamma\left(\frac{d+1}{2}\right)}{\Gamma\left(\frac{d}{2}\right)\sqrt{d\pi}} \int_{\mathbb{R}} \frac{\left(1 + \frac{(y+a)^2}{d}\right)^{\frac{d+1}{2}}}{\left(1 + \frac{y^2}{d}\right)^{d+1}} dy - 1.
$$

this is a polynomial of degree $d$ in the variable $a$. We have the following Lemma proven in Section B.3.4.

**Lemma 10** *For $a \geq 0$ and $d \geq 0$, we have the following algebraic inequality.*

$$\int_{\mathbb{R}} \frac{\left(1 + \frac{(y+a)^2}{d}\right)^{\frac{d+1}{2}}}{\left(1 + \frac{y^2}{d}\right)^{d+1}} \mathrm{d}y \leq \frac{a^2}{2\sqrt{d}}(d+1)^2 \left(2 + \frac{a}{\sqrt{d}}\right)^{d-1} + \int_{\mathbb{R}} \frac{(1 + y^2/d)^{\frac{d+1}{2}}}{\left(1 + \frac{y^2}{d}\right)^{d+1}} \mathrm{d}y.$$

Using this lemma, and because we recognize up to a constant the integral of the student distribution on $\mathbb{R}$ in the right-hand side, we have

$$d_{\chi^2}(f_a, f_0) = \frac{\Gamma\left(\frac{d+1}{2}\right)}{\Gamma\left(\frac{d}{2}\right)\sqrt{d\pi}} \left(\frac{a^2}{2\sqrt{d}}(d+1)^2 \left(2 + \frac{a}{\sqrt{d}}\right)^{d-1} + \int_{\mathbb{R}} \frac{(1 + y^2/d)^{\frac{d+1}{2}}}{\left(1 + \frac{y^2}{d}\right)^{d+1}} \mathrm{d}y\right) - 1$$

$$\leq \frac{\Gamma\left(\frac{d+1}{2}\right)}{\Gamma\left(\frac{d}{2}\right)\sqrt{d\pi}} \frac{a^2}{2\sqrt{d}}(d+1)^2 \left(2 + \frac{a}{\sqrt{d}}\right)^{d-1}$$

then, use that for any $d \geq 1$, $\Gamma(\frac{d+1}{2}) \leq \Gamma(\frac{d}{2})\sqrt{d/2}$ from Wendel (1948), hence

$$d_{\chi^2}(f_a, f_0) \leq \frac{a^2(d+1)^2}{2\sqrt{2d\pi}} \left(2 + \frac{a}{\sqrt{d}}\right)^{d-1} \leq \frac{a^2(d+1)^2}{5\sqrt{d}} \left(2 + \frac{a}{\sqrt{d}}\right)^{d-1},$$

using $2\sqrt{2\pi} \geq 5 \cdot$ Then, we use the link between KL divergence and $\chi^2$ divergence to get the result.

$$D_{\mathrm{KL}}(f_a, f_0) \leq \log(1 + d_{\chi^2}(f_a, f_0))$$

$$\leq \log\left(1 + \frac{a^2(d+1)^2}{5\sqrt{d}} \left(2 + \frac{a}{\sqrt{d}}\right)^{d-1}\right) \tag{30}$$

Then, we have,

$$\log\left(1 + \frac{a^2(d+1)^2}{5\sqrt{d}} \left(2 + \frac{a}{\sqrt{d}}\right)^{d-1}\right) \leq \begin{cases} \log\left(1 + 3^{d-1}\frac{(d+1)^2}{5\sqrt{d}}a^2\right) & \text{if } a < 1 \\ \log\left(1 + \frac{(d+1)^2}{5\sqrt{d}}a^{d+1}\left(\frac{(d+1)^2}{\sqrt{d}} + \frac{1}{\sqrt{d}}\right)^{d-1}\right) & \text{if } a \geq 1 \end{cases}$$

hence, using that $1 \leq 3^{d-1}\frac{(d+1)^2}{d}a^{d+1}$

$$\log\left(1 + \frac{a^2(d+1)^2}{d} \left(2 + \frac{a}{\sqrt{d}}\right)^{d-1}\right) \leq \begin{cases} 3^{d-1}\frac{(d+1)^2}{5\sqrt{d}}a^2 & \text{if } a < 1 \\ (d+1)\log(a) + \log\left(3^d\frac{(d+1)^2}{5\sqrt{d}}\right) & \text{if } a \geq 1. \end{cases}$$

Inject this in Equation (30) to get the result.

### B.1.3   Proof of Lemma 4: KL for Corrupted Bernoulli Distribution

Let $\alpha \in (0, 1/2)$ and denote $\delta_x$ the Dirac distribution in $x$. Define
$P_0 = (1 - \alpha)\delta_0 + \alpha\delta_1,$
$P_1 = \alpha\delta_0 + (1 - \alpha)\delta_1,$
$Q_0 = (1 - \varepsilon)(1 - \alpha)\delta_0 + (1 - (1 - \varepsilon)(1 - \alpha))\delta_1,$
$Q_1 = (1 - (1 - \varepsilon)(1 - \alpha))\delta_0 + (1 - \varepsilon)(1 - \alpha)\delta_1.$

One can check that $Q_0 = (1-\varepsilon)P_0 + \varepsilon\delta_1$ and $Q_1 = (1-\varepsilon)P_1 + \varepsilon\delta_0$ and hence $Q_0$ and $Q_1$ are in the $\varepsilon$-corrupted neighborhood of respectively $P_0$ and $P_1$.

We have

$$
\begin{aligned}
D_{\mathrm{KL}}(Q_0, Q_1) &= \sum_{k \in \{0, c\}} \mathbb{P}_{Q_0}(X = k) \log \left( \frac{\mathbb{P}_{Q_0}(X = k)}{\mathbb{P}_{Q_1}(X = k)} \right) \\
&= (1 - \varepsilon)(1 - \alpha) \log \left( \frac{(1 - \varepsilon)(1 - \alpha)}{1 - (1 - \varepsilon)(1 - \alpha)} \right) + (1 - (1 - \varepsilon)(1 - \alpha)) \log \left( \frac{1 - (1 - \varepsilon)(1 - \alpha)}{(1 - \varepsilon)(1 - \alpha)} \right) \\
&= ((1 - \varepsilon)(1 - \alpha) - (1 - (1 - \varepsilon)(1 - \alpha))) \log \left( \frac{(1 - \varepsilon)(1 - \alpha)}{1 - (1 - \varepsilon)(1 - \alpha)} \right) \\
&= (1 - 2\varepsilon - 2\alpha + 2\varepsilon\alpha) \log \left( 1 + \frac{1 - 2\varepsilon - 2\alpha + 2\varepsilon\alpha}{\varepsilon + \alpha - \varepsilon\alpha} \right)
\end{aligned}
$$

Then, note that $\Delta = \mathbb{E}_{P_1}[X] - \mathbb{E}_{P_0}[X] = (1 - 2\alpha)$ and $\sigma^2 = \mathrm{Var}_{P_0}(X) = \mathrm{Var}_{P_1}(X) = \alpha(1 - \alpha)$. Hence, with $\alpha = \frac{1}{2}(1 - \Delta)$.

$$
D_{\mathrm{KL}}(Q_0, Q_1) = (1 - 2\varepsilon - (1 - \Delta)(1 - \varepsilon)) \log \left( 1 + \frac{1 - 2\varepsilon - (1 - \Delta)(1 - \varepsilon)}{\varepsilon + \frac{1}{2}(1 - \Delta)(1 - \varepsilon)} \right) \tag{31}
$$

$$
= (\Delta(1 - \varepsilon) - \varepsilon) \log \left( 1 + \frac{\Delta(1 - \varepsilon) - \varepsilon}{\frac{1}{2}(1 + \varepsilon) - \frac{1}{2}\Delta(1 - \varepsilon)} \right) \tag{32}
$$

**Uniform bound**: if $\varepsilon > 0$, we have

$$
D_{\mathrm{KL}}(Q_0, Q_1) \leq (1 - 2\varepsilon) \log \left( 1 + \frac{1 - 2\varepsilon}{\varepsilon} \right).
$$

**High distinguishibility regime**: in the setting $2\sigma > \Delta$, we have the bound

$$
\begin{aligned}
D_{\mathrm{KL}}(Q_0, Q_1) &\leq \left( \frac{\Delta}{2\sigma}(1 - \varepsilon) - \varepsilon \right) \log \left( 1 + 2 \frac{\frac{\Delta}{2\sigma}(1 - \varepsilon) - \varepsilon}{1 - \left( \frac{\Delta}{2\sigma}(1 - \varepsilon) - \varepsilon \right)} \right) \\
&= \left( \frac{\Delta(1 - \varepsilon) - 2\sigma\varepsilon}{2\sigma} \right) \log \left( 1 + 2 \frac{\Delta(1 - \varepsilon) - 2\sigma\varepsilon}{2\sigma - (\Delta(1 - \varepsilon) - 2\sigma\varepsilon)} \right)
\end{aligned}
$$

**Low distinguishibility regime**: if $\Delta \leq 2\sigma \frac{\varepsilon}{\sqrt{1 - 2\varepsilon}}$. Then there exists $\varepsilon' \leq \varepsilon$ such that $\Delta = 2\sigma \frac{\varepsilon'}{\sqrt{1 - 2\varepsilon'}}$ and then, from Equation (31), there exists $Q_0', Q_1'$ which are $\varepsilon'$-corrupted versions of $P_0$ and $P_1$ such that $KL(Q_0', Q_1') = 0$

### B.2 Lemmas for Regret upper bound

#### B.2.1 Proof of Corollary 1: Simplified Upper Bound of HuberUCB

Replacing $\beta_i$ by $4\sigma_i$, we have

• If $\widetilde{\Delta}_{i,\varepsilon} > 6\sigma_i \left(1 + 4\sqrt{2}\overline{\varepsilon}\right)^2$, then

$$\mathbb{E}[T_i(n)] \leq \log(n) \max\left(\frac{128\sigma_i}{3\widetilde{\Delta}_{i,\varepsilon}}, \frac{4}{(p-5\varepsilon)^2}\left(1 + 2\sqrt{2}\left(\overline{\varepsilon} \vee \frac{9}{14\sqrt{2}}\right)\right)^2\right) + 10(\log(n) + 1)$$

• If $\widetilde{\Delta}_{i,\varepsilon} > 6\sigma_i \left(1 + 4\sqrt{2}\overline{\varepsilon}\right)^2$, then

$$\mathbb{E}[T_i(n)] \leq \log(n) \max\left(\frac{50\sigma_i^2}{9\widetilde{\Delta}_{i,\varepsilon}^2}\left(\sqrt{2} + 8\overline{\varepsilon}\right)^2, \frac{4}{(p-5\varepsilon)^2}\left(1 + 2\sqrt{2}\left(\overline{\varepsilon} \vee \frac{9}{14\sqrt{2}}\right)\right)^2\right) + 10(\log(n) + 1).$$

Then, we use that

$$\left(1 + 2\sqrt{2}\left(\overline{\varepsilon} \vee \frac{9}{14\sqrt{2}}\right)\right)^2 \leq 2\left(1 + \left(2\sqrt{2}\left(\overline{\varepsilon} \vee \frac{9}{14\sqrt{2}}\right)\right)^2\right)$$

$$= 2 + 8\left(\overline{\varepsilon}^2 \vee \frac{81}{392}\right) \leq 8\overline{\varepsilon}^2 + 2 + \frac{648}{392} \leq 8\overline{\varepsilon}^2 + 4$$

and that $p - 5\varepsilon \geq 1/4$, to get

• If $\widetilde{\Delta}_{i,\varepsilon} > 6\sigma_i \left(1 + 4\sqrt{2}\overline{\varepsilon}\right)^2$, then

$$\mathbb{E}[T_i(n)] \leq \log(n) \max\left(\frac{128\sigma_i}{3\widetilde{\Delta}_{i,\varepsilon}}, 512\overline{\varepsilon}^2 + 256\right) + 10(\log(n) + 1)$$

$$= \frac{128}{3} \log(n) \max\left(\frac{\sigma_i}{\widetilde{\Delta}_{i,\varepsilon}}, 12\overline{\varepsilon}^2 + 6\right) + 10(\log(n) + 1)$$

$$\leq 43 \log(n) \max\left(\frac{\sigma_i}{\widetilde{\Delta}_{i,\varepsilon}}, 12\overline{\varepsilon}^2 + 6\right) + 10(\log(n) + 1)$$

• If $\widetilde{\Delta}_{i,\varepsilon} > 6\sigma_i \left(1 + 4\sqrt{2}\overline{\varepsilon}\right)^2$, then

$$\mathbb{E}[T_i(n)] \leq \log(n) \max\left(\frac{50\sigma_i^2}{9\widetilde{\Delta}_{i,\varepsilon}^2}\left(\sqrt{2} + 8\overline{\varepsilon}\right)^2, 512\overline{\varepsilon}^2 + 256\right) + 10(\log(n) + 1)$$

$$\leq \log(n) \max\left(\frac{100\sigma_i^2}{9\widetilde{\Delta}_{i,\varepsilon}^2}\left(2 + 64\overline{\varepsilon}^2\right), 512\overline{\varepsilon}^2 + 256\right) + 10(\log(n) + 1)$$

$$\leq 23 \log(n) \max\left(\frac{\sigma_i^2}{\widetilde{\Delta}_{i,\varepsilon}^2}\left(1 + 32\overline{\varepsilon}^2\right), 24\overline{\varepsilon}^2 + 12\right) + 10(\log(n) + 1)$$

#### B.2.2 Proof of Lemma 6: Regret Upper bound for SeqHuberUCB

In this section we virtually copy the proof of the regret for HuberUCB done in Section A.4 with modified constants and using the crude bound $P_2(s) \geq s/2$ whenever necessary.

If $A_t = i$ then at least one of the following four inequalities is true:

$$\widehat{\text{SeqHub}}_{1,T_1(t-1)} + B_1(T_1(t-1), t) \leq \mu_1 \tag{33}$$

or

$$\widehat{\text{SeqHub}}_{i,T_i(t-1)} \geq \mu_i + B_i(T_i(t-1), t) \tag{34}$$

or

$$\Delta_i < 2B_i(T_i(t-1), t) \tag{35}$$

or

$$P_2(T_1(t-1)) < s_{lim}(t) = \frac{98 \log(t)}{128 \left(p - 5\varepsilon\right)^2} \left(1 + 2\sqrt{2}\left(\bar{\varepsilon} \vee \frac{9}{14\sqrt{2}}\right)\right)^2 \tag{36}$$

Indeed, if $P_2(T_i(t-1)) < s_{lim}(t)$, then $B_i(T_i(t-1), t) = \infty$ and Inequality (35) is true. On the other hand, if $P_2(T_i(t-1)) \geq s_{lim}(t)$, then we have $B_i(T_i(t-1), t)$ is finite and all four inequalities are false, then,

$$\widehat{\text{SeqHub}}_{1,T_1(t-1)} + B_1(T_1(t-1), t) > \mu_1$$
$$= \mu_i + \Delta_i$$
$$\geq \mu_i + 2B_i(T_i(t-1), n)$$
$$\geq \mu_i + 2B_i(T_i(t-1), t)$$
$$\geq \widehat{\text{SeqHub}}_{i,T_i(t-1)} + B_i(T_i(t-1), t)$$

which implies that $A_t \neq i$.

**Step 1.** We have that $\mathbb{P}\left((33) \text{ is true}\right) \leq 14/t$.
  PROOF:

Then, we have that,

$$\mathbb{P}\left(\widehat{\text{SeqHub}}_{1,T_1(t-1)} + B_1(T_1(t-1), t) \leq \mu_1\right) \leq \sum_{s=1}^{t} \mathbb{P}\left(\widehat{\text{SeqHub}}_{1,s} + B_1(s, t) \leq \mu_1\right)$$
$$= \sum_{s=\lceil s_{lim}(t)\rceil}^{t} \mathbb{P}\left(\widehat{\text{SeqHub}}_{1,s} - \mu_1 \leq -B_1(s, t)\right)$$

Then, use Theorem 4, we get

$$\mathbb{P}\left(\widehat{\text{SeqHub}}_{1,T_1(t-1)} + B_1(T_1(t-1), t) \leq \mu_1\right) \leq \sum_{s=\lceil s_{lim}(t)\rceil}^{t} 14 e^{-\log(t^2)}$$
$$\leq \sum_{s=\lceil s_{lim}(t)\rceil}^{t} \frac{14}{t^2} \leq \frac{14}{t}.$$

**Step 2.** Similarly, for arm $i$, we have

$$\mathbb{P}\left(\widehat{\text{SeqHub}}_{i,T_i(t-1)} \geq \mu_i + B_i(T_i(t-1), t)\right) \leq \frac{14}{t}$$

  PROOF:   We have,

$$\mathbb{P}\left(\widehat{\text{SeqHub}}_{i,T_i(t-1)} \geq \mu_i + B_i(T_i(t-1), t)\right) \leq \sum_{s=\lceil s_{lim}(t)\rceil}^{t} \mathbb{P}\left(\widehat{\text{SeqHub}}_{i,s} - \mu_i \geq B_i(s, t)\right)$$
$$\leq \sum_{s=\lceil s_{lim}(t)\rceil}^{t} 14 e^{-\log(t^2)} \leq \frac{14}{t}.$$

**Step 3.** Let $v \in \mathbb{N}$. If one of the two following conditions are true, then for all $t$ such that $P_2(T_i(t-1)) \geq v$, we have $\Delta_i \geq 2B_i(T_i(t-1), t)$ (i.e. Equation (35) is false).

Condition 1: if $\widetilde{\Delta}_{i,\varepsilon} > 12\frac{\sigma_i^2}{\beta_i}\left(\sqrt{2} + 2\frac{\beta_i}{\sigma_i}\overline{\varepsilon}\right)^2$ and $v \leq \log(t)\frac{96\beta_i}{9\widetilde{\Delta}_{i,\varepsilon}}$.

Condition 2: if $\widetilde{\Delta}_{i,\varepsilon} \leq 12\frac{\sigma_i^2}{\beta_i}\left(\sqrt{2} + 2\frac{\beta_i}{\sigma_i}\overline{\varepsilon}\right)^2$ and $v \leq \frac{50}{9\widetilde{\Delta}_{i,\varepsilon}^2}\left(\sigma_i\sqrt{2} + 2\beta_i\overline{\varepsilon}\right)^2\log(t)$.

PROOF: We search for the smallest value $v \geq s_{lim}(t)$ such that $\Delta_i$ verifies

$$\Delta_i \geq 2B_i(v, t) = 2r_v(1/t^2) + 2\left(\frac{1}{p - \sqrt{\frac{\log(t^2)}{2v}} - \varepsilon} - 1\right)r_{P_2(v)}(1/t^2) + 2b_i.$$

First, we simplify the expression, having that $v \geq s_{lim}(t)$, we have

$$\frac{\log(t^2)}{2v} \leq \frac{128(p - 5\varepsilon)^2}{98(1 + 9/7)^2} \leq \frac{(p - \varepsilon)^2}{4},$$

hence $r_v(1/t^2) \leq \frac{2}{(p-\varepsilon)}\left(\sigma_i\sqrt{\frac{2\log(t^2)}{v}}\right)$ and we simplify the condition to

$$\Delta_i \geq \frac{4}{(p - \varepsilon)}\left(\sigma_i\sqrt{\frac{2\log(t^2)}{v}} + \beta_i\frac{\log(t^2)}{3v} + 2\beta_i\overline{\varepsilon}\sqrt{\frac{\log(t^2)}{v}} + 2\beta_i\varepsilon\right)$$

$$+ \frac{4}{p - \varepsilon}\left(\sigma_i\sqrt{\frac{2\log(t^2)}{P_2(v)}} + \beta_i\frac{\log(t^2)}{3P_2(v)} + 2\beta_i\overline{\varepsilon}\sqrt{\frac{\log(t^2)}{P_2(v)}} + 2\beta_i\varepsilon\right) + 2b_i$$

$$\geq \frac{12}{(p - \varepsilon)}\left(\sigma_i\sqrt{\frac{2\log(t^2)}{v}} + \beta_i\frac{\log(t^2)}{3v} + 2\beta_i\overline{\varepsilon}\sqrt{\frac{\log(t^2)}{v}} + 2\beta_i\varepsilon\right) + 2b_i$$

where we used that $P_2(v) \geq v/2$.

Let us denote $\widetilde{\Delta}_{i,\varepsilon} = (\Delta_i - 2b_i)(p - \varepsilon) - 24\beta_i\varepsilon$, we are searching for $v$ such that

$$\beta_i\frac{\log(t^2)}{3v} + \sqrt{\frac{\log(t^2)}{v}}\left(\sigma_i\sqrt{2} + 2\beta_i\overline{\varepsilon}\right) - \frac{\widetilde{\Delta}_{i,\varepsilon}}{12} \leq 0$$

This is a second order polynomial in $\sqrt{\log(t^2)/v}$.

If $\widetilde{\Delta}_{i,\varepsilon} > 0$, then the smallest $v > 0$ is

$$\sqrt{\frac{\log(t^2)}{v}} = \frac{3}{2\beta_i}\left(-\left(\sigma_i\sqrt{2} + 2\overline{\varepsilon}\beta_i\right) + \sqrt{\left(\sigma_i\sqrt{2} + 2\beta_i\overline{\varepsilon}\right)^2 + \frac{\widetilde{\Delta}_{i,\varepsilon}\beta_i}{9}}\right).$$

**First setting:** if $\widetilde{\Delta}_{i,\varepsilon} > 36\frac{\sigma_i^2}{\beta_i}\left(\sqrt{2} + 2\frac{\beta_i}{\sigma_i}\overline{\varepsilon}\right)^2$,

In that case, we have

$$\sqrt{\frac{\log(t^2)}{v}} \geq \frac{3}{2\beta_i}\left(-\left(\sigma_i\sqrt{2} + 2\beta_i\overline{\varepsilon}\right) + \sqrt{\frac{\beta_i\widetilde{\Delta}_{i,\varepsilon}}{9}}\right) \geq \frac{3}{2\beta_i}\sqrt{\frac{\beta_i\widetilde{\Delta}_{i,\varepsilon}}{36}} = \sqrt{\frac{\widetilde{\Delta}_{i,\varepsilon}}{16\beta_i}}$$

Hence, $v \leq \log(t)\frac{32\beta_i}{\widetilde{\Delta}_{i,\varepsilon}}$.

**Second setting:** if $\widetilde{\Delta}_{i,\varepsilon} \leq 36\frac{\sigma_i^2}{\beta_i}\left(\sqrt{2} + 2\frac{\beta_i}{\sigma_i}\overline{\varepsilon}\right)^2$, then we use Lemma 9, using that

$$\frac{\widetilde{\Delta}_{i,\varepsilon}\beta_i}{9\left(\sigma_i\sqrt{2} + 2\beta_i\overline{\varepsilon}\right)^2} \leq 4$$

and the fact that $\frac{\sqrt{1+4}-1}{4} \geq \frac{3}{10}$, we get,

$$\sqrt{\frac{\log(t^2)}{v}} \geq \frac{\widetilde{\Delta}_{i,\varepsilon}}{20\left(\sigma_i\sqrt{2} + 2\beta_i\overline{\varepsilon}\right)}$$

Hence,

$$v \leq \frac{40}{\widetilde{\Delta}_{i,\varepsilon}^2}\left(\sigma_i\sqrt{2} + 2\beta_i\overline{\varepsilon}\right)^2 \log(t).$$

**Step 4.** Using All the previous steps, we prove the theorem.

PROOF: We have

$$\mathbb{E}[T_i(t)] = \mathbb{E}\left[\sum_{t=1}^{t}\mathbf{1}\{A_t = i\}\right]$$

$$\leq \lfloor\max(v, 2s_{lim}(t))\rfloor + \mathbb{E}\left[\sum_{t=\lfloor\max(v, 2s_{lim}(t))\rfloor+1}^{t}\mathbf{1}\{A_t = i \text{ and } (35) \text{ is false}\}\right]$$

$$\leq \lfloor\max(v, 2s_{lim}(t))\rfloor + \mathbb{E}\left[\sum_{t=\lfloor\max(v, 2s_{lim}(t))\rfloor+1}^{t}\mathbf{1}\{(33) \text{ or } (34) \text{ or } (36) \text{ is true}\}\right]$$

$$= \lfloor\max(v, 2s_{lim}(t))\rfloor + \sum_{t=\lfloor\min(v, 2s_{lim}(t))\rfloor+1}^{t}\mathbb{P}\left((33) \text{ or } (34) \text{ is true}\right)$$

$$\leq \lfloor\max(v, 2s_{lim}(t))\rfloor + 2\sum_{t=\lfloor\min(v, 2s_{lim}(t))\rfloor+1}^{t}\frac{14}{t}$$

using the harmonic series bound by $\log(t) + 1$, we have

$$\mathbb{E}[T_i(t)] \leq \max(v, 2s_{lim}(t)) + 28(\log(t) + 1)$$

Then, we replace the value of $v$,

**First setting:** $\widetilde{\Delta}_{i,\varepsilon} > 36\frac{\sigma_i^2}{\beta_i}\left(\sqrt{2} + 2\frac{\beta_i}{\sigma_i}\overline{\varepsilon}\right)^2$

$$\mathbb{E}[T_i(t)] \leq \log(t)\max\left(\frac{32\beta_i}{\widetilde{\Delta}_{i,\varepsilon}}, \frac{8}{(p-5\varepsilon)^2}\left(1 + 2\sqrt{2}\left(\overline{\varepsilon} \vee \frac{9}{14\sqrt{2}}\right)\right)^2\right) + 28(\log(t) + 1)$$

**Second setting:** if $\widetilde{\Delta}_{i,\varepsilon} \leq 36\frac{\sigma_i^2}{\beta_i}\left(\sqrt{2} + 2\frac{\beta_i}{\sigma_i}\overline{\varepsilon}\right)^2$, then

$$\mathbb{E}[T_i(t)] \leq \log(n)\max\left(\frac{40}{\widetilde{\Delta}_{i,\varepsilon}^2}\left(\sigma_i\sqrt{2} + 2\beta_i\overline{\varepsilon}\right)^2, \frac{8}{(p-5\varepsilon)^2}\left(1 + 2\sqrt{2}\left(\overline{\varepsilon} \vee \frac{9}{14\sqrt{2}}\right)\right)^2\right) + 28(\log(t) + 1).$$

Finish the proof of the Theorem using the given values for the constants $\beta_i, \varepsilon, p$.

### B.3 Lemmas for concentration of robust estimators

#### B.3.1 Proof of Lemma 7: Controlling Variance of Influence of Huber's Estimator

Let $\rho_\beta$ be Huber's loss function, with $\psi_\beta = \rho'_\beta$. We have that for any $x > 0$, $\psi_\beta(x)^2 \leq 2\rho_\beta(x)$. Hence,

$$\mathrm{Var}(\psi_\beta(Y - \mathrm{Hub}_\beta(P))) = \mathbb{E}[\psi_\beta(Y - \mathrm{Hub}_\beta(P))^2] \leq 2\mathbb{E}[\rho_\beta(Y - \mathrm{Hub}_\beta(P))].$$

Then, use that by definition of $\mathrm{Hub}_\beta(P)$, $\mathrm{Hub}_\beta(P)$ is a minimizer of $\theta \mapsto \mathbb{E}[\rho_\beta(Y - \theta)]$, hence,

$$\mathrm{Var}(\psi_\beta(Y - \mathrm{Hub}_\beta(P))) \leq 2\mathbb{E}[\rho_\beta(Y - \mathbb{E}[Y])].$$

and finally, use that $\rho_\beta(x) \leq x^2/2$ to conclude.

#### B.3.2 Proof of Lemma 8 : Concentrating Huber's Estimator by Concentrating the Influence

For all $n \in \mathbb{N}^*$, $\lambda > 0$, let

$$f_n(\lambda) = \frac{\mathrm{sign}(\Delta_n)}{n} \sum_{i=1}^n \psi_\beta(X_i - \mathrm{Hub}_\beta(P) - \lambda\,\mathrm{sign}(\Delta_n)),$$

where $\Delta_n = \mathrm{Hub}_\beta(P) - \mathrm{Hub}_\beta(X_1^n)$.

**Step 1.** For any $\lambda > 0$, $\mathbb{P}(|\Delta_n| \geq \lambda) \leq \mathbb{P}(f_n(\lambda) \geq 0)$.
PROOF: For all $y \in \mathbb{R}$, let $J_n(y) = \frac{1}{n} \sum_{i=1}^n \rho_\beta(X_i - y)$ we have,

$$J_n''(y) = \frac{1}{n} \sum_{i=1}^n \psi'_\beta(X_i - y).$$

In particular, having $f_n(\lambda) = -\mathrm{sign}(\Delta_n)J'(\mathrm{Hub}_\beta(P) + \lambda\,\mathrm{sign}(\Delta_n))$ if we take the derivative of $f_n$ with respect to $\lambda$, we have the following equation

$$\frac{\partial}{\partial\lambda} f_n(\lambda) = -\mathrm{sign}(\Delta_n)^2 J_n''(\mathrm{Hub}_\beta(P) + \lambda\,\mathrm{sign}(\Delta_n))$$

$$\leq -\frac{1}{n} \sum_{i=1}^n \psi'_\beta(X_i - \mathrm{Hub}_\beta(P) - \lambda\,\mathrm{sign}(\Delta_n)). \tag{37}$$

Then, because $\psi'_\beta$ is non-negative, the function $\lambda \mapsto f_n(\lambda,)$ is non-increasing. Hence, for all $n \in \mathbb{N}^*$ and $\lambda > 0$,

$$|\Delta_n| \geq \lambda \Rightarrow f_n(|\Delta_n|) = 0 \leq f_n(\lambda),$$

Hence,

$$\mathbb{P}(|\Delta_n| \geq \lambda) \leq \mathbb{P}(f_n(\lambda) \geq 0). \tag{38}$$

**Step 2.** For all $\lambda > 0$,

$$f_n(\lambda) \leq f_n(0) - \lambda \inf_{t \in [0,\lambda]} |f'_n(t)|.$$

PROOF: We apply Taylor's inequality to the function $f_n$. As $f_n$ is non-increasing (because its derivative is non-positive, see Equation (37)), we get

$$f_n(\lambda) \leq f_n(0) - \lambda \inf_{t \in [0,\lambda]} |f'_n(t)|.$$

**Step 3.** Let $m_n = \mathbb{E}\left[\inf_{t\in[0,\lambda]} \frac{1}{n}\sum_{i=1}^n \psi'_\beta(X'_i - \mathrm{Hub}_\beta(P) - t)\right]$. With probability larger than $1 - 2e^{-2n\eta^2}$,

$$\inf_{t\in[0,\lambda]} |f'_n(t)| \geq m_n - 2\eta - \varepsilon,$$

PROOF:  Write that $X_i = (1 - W_i)Y_i + W_i Z_i$ where $W_1, \ldots, W_n$ are i.i.d Bernoulli random variable with mean $\varepsilon$, $Y_1, \ldots, Y_n$ are i.i.d $\sim P$ and $Z_1, \ldots, Z_n$ are i.i.d with law $H$.

From equation (37),

$$\begin{aligned}
|f'_n(t))| \geq & \frac{1}{n}\sum_{i=1}^n \psi'_\beta(X_i - \mathrm{Hub}_\beta(P) - t\,\mathrm{sign}(\Delta)) \\
\geq & \frac{1}{n}\sum_{i=1}^n \mathbf{1}\{W_i = 0\}\psi'_\beta(Y_i - \mathrm{Hub}_\beta(P) - t\,\mathrm{sign}(\Delta)) & (39) \\
& + \frac{1}{n}\sum_{i=1}^n \mathbf{1}\{W_i = 1\}\psi'_\beta(Z_i - \mathrm{Hub}_\beta(P) - t\,\mathrm{sign}(\Delta)) & (40) \\
\geq & \frac{1}{n}\sum_{i=1}^n \psi'_\beta(Y_i - \mathrm{Hub}_\beta(P) - t\,\mathrm{sign}(\Delta)) & (41) \\
& + \frac{1}{n}\sum_{i=1}^n \mathbf{1}\{W_i = 1\}\left(\psi'_\beta(Z_i - \mathrm{Hub}_\beta(P) - t\,\mathrm{sign}(\Delta)) - \psi'_\beta(W_i - \mathrm{Hub}_\beta(P) - t\,\mathrm{sign}(\Delta))\right) & (42)
\end{aligned}$$

Hence, because $\psi'_\beta \in [0,1]$, we have

$$|f'_n(t))| \geq \frac{1}{n}\sum_{i=1}^n \psi'_\beta(Y_i - \mathrm{Hub}_\beta(P) - t\,\mathrm{sign}(\Delta)) - \frac{1}{n}\sum_{i=1}^n \mathbf{1}\{W_i = 1\}) \tag{43}$$

The right-hand side depends on the infimum of the mean of $n$ i.i.d random variables in $[0,1]$. Hence, the function

$$Z(X_1^n) \mapsto \sup_{t\in[0,\lambda]} \sum_{i=1}^n \psi'_\beta(X'_i - \mathrm{Hub}_\beta(P) - t)$$

satisfies, by sub-linearity of the supremum operator and triangular inequality, the bounded difference property, with differences bounded by 1. Hence, by Hoeffding's inequality, we get with probability larger than $1 - e^{-2n\eta^2}$,

$$\inf_{t\in[0,\lambda]} |f'_n(t))| \geq \mathbb{E}\left[\inf_{t\in[0,\lambda]} \frac{1}{n}\sum_{i=1}^n \psi'_\beta(X'_i - \mathrm{Hub}_\beta(P) - t)\right] - \eta - \frac{1}{n}\sum_{i=1}^n \mathbf{1}\{W_i = 1\})$$

and using Hoeffding's inequality to control $\frac{1}{n}\sum_{i=1}^n \mathbf{1}\{W_i = 1\}$, we have with probability larger than $1 - 2e^{-2\eta^2/n}$,

$$\inf_{t\in[0,\lambda]} |f'_n(t))| \geq \mathbb{E}\left[\inf_{t\in[0,\lambda]} \frac{1}{n}\sum_{i=1}^n \psi'_\beta(X'_i - \mathrm{Hub}_\beta(P) - t)\right] - 2\eta - \varepsilon$$

**Step 4.** For $\lambda \in (0, \beta/2)$,

$$\mathbb{P}\left(\ |\Delta_n| \geq \lambda\right) \leq \mathbb{P}\left(\left|\frac{1}{n}\sum_{i=1}^n \psi_\beta(X_i - \mathrm{Hub}_\beta(P))\right| \geq \lambda\left(m_n - \eta - \varepsilon\right)\right) + 2e^{-2n\eta^2}.$$

PROOF: For any $\lambda > 0$, we have

$$\mathbb{P}(|\Delta_n| \geq \lambda) \leq \mathbb{P}(f_n(\lambda) \geq 0) \qquad\qquad \text{(from Step 1)}$$

$$\leq 1 - \mathbb{P}\left(f_n(0) - \lambda \inf_{t\in[0,\lambda]} |f_n'(t)| \leq 0\right) \qquad\qquad \text{(from Step 2)}$$

$$\leq 1 - \mathbb{P}\left(f_n(0) \leq \lambda\left(m_n - 2\eta - \varepsilon\right)\right) + 2e^{-2n\eta^2} \qquad\qquad \text{(from Step 3)}$$

$$= \mathbb{P}\left(\left|\frac{1}{n}\sum_{i=1}^{n} \psi_\beta(X_i - \mathrm{Hub}_\beta(P))\right| \geq \lambda\left(m_n - \eta - \varepsilon\right)\right) + 2e^{-2n\eta^2}. \qquad\qquad (44)$$

**Step 5.** We prove that $m_n \geq p$, and hence

$$\mathbb{P}\left(|\Delta_n| \geq \lambda\right) \leq \mathbb{P}\left(\left|\frac{1}{n}\sum_{i=1}^{n} \psi_\beta(X_i - \mathrm{Hub}(P))\right| \geq \lambda\left(p - \eta - \varepsilon\right)\right) + 2e^{-2n\eta^2}$$

PROOF: For all $\lambda \leq \beta/2$,

$$\mathbb{E}\left[\inf_{t\in[0,\lambda]} \frac{1}{n}\sum_{i=1}^{n} \psi_\beta'(X_i' - \mathrm{Hub}_\beta(P) - t)\right] = \mathbb{E}\left[\inf_{t\in[0,\lambda]} \frac{1}{n}\sum_{i=1}^{n} \mathbf{1}\{|X_i' - \mathrm{Hub}_\beta(P) - t| \leq \beta\}\right]$$

$$\geq \mathbb{E}\left[\frac{1}{n}\sum_{i=1}^{n} \mathbf{1}\{|X_i' - \mathrm{Hub}_\beta(P)| \leq \beta - \lambda\}\right]$$

$$\geq \mathbb{E}\left[\frac{1}{n}\sum_{i=1}^{n} \mathbf{1}\{|X_i' - \mathrm{Hub}_\beta(P)| \leq \beta/2\}\right] = p$$

Then, we plug the bound on $m_n$ found in the previous step in equation (44), we get for any $\eta > 0$ and $\lambda \in (0, \beta/2]$,

$$\mathbb{P}(|\Delta_n| \geq \lambda) \leq \mathbb{P}\left(\left|\frac{1}{n}\sum_{i=1}^{n} \psi_\beta(X_i - \mathrm{Hub}_\beta(P))\right| \geq \lambda\left(p - \eta - \varepsilon\right)\right) + 2e^{-2n\eta^2}$$

### B.3.3 Proof of Lemma 9: Algebra tool for bounding polinomial roots

The solutions of the second order polynomial indicate that $x$ must verify

$$x \geq \frac{-b + \sqrt{b^2 + 4ac}}{2a} \geq \frac{b}{2a}\left(-1 + \sqrt{1 + \frac{4ac}{b^2}}\right).$$

Then, use that the function $x \mapsto \sqrt{x+1}$ is concave and hence the graph of $x \mapsto \sqrt{x+1}$ is above its chords and we have for any $x \in [0, d]$, $\sqrt{1+x} \geq 1 + x\frac{\sqrt{d+1}-1}{d}$. Hence,

$$x \geq \frac{b}{2a}\left(\frac{4ac(\sqrt{d+1}-1)}{db^2}\right) = \frac{2c(\sqrt{d+1}-1)}{db}.$$

### B.3.4 Proof of Lemma 10: Algebra on Student's distribution

We have,

$$
\int_{\mathbb{R}} \frac{\left(1 + \frac{(y+a)^2}{d}\right)^{\frac{d+1}{2}}}{\left(1 + \frac{y^2}{d}\right)^{d+1}} \mathrm{d}y = \int_{\mathbb{R}} \sum_{l=0}^{\frac{d+1}{2}} \binom{\frac{d+1}{2}}{l} \frac{(y+a)^{2l}}{d^l \left(1 + \frac{y^2}{d}\right)^{d+1}} \mathrm{d}y
$$

$$
= \int_{\mathbb{R}} \sum_{l=0}^{\frac{d+1}{2}} \sum_{j=0}^{2l} \binom{\frac{d+1}{2}}{l} \binom{2l}{j} \frac{y^j a^{2l-j}}{d^l \left(1 + \frac{y^2}{d}\right)^{d+1}} \mathrm{d}y
$$

$$
= \sum_{l=0}^{\frac{d+1}{2}} \sum_{j=0}^{2l} \binom{\frac{d+1}{2}}{l} \binom{2l}{j} \int_{\mathbb{R}} \frac{y^j a^{2l-j}}{d^l \left(1 + \frac{y^2}{d}\right)^{d+1}} \mathrm{d}y
$$

Remark that the integral is 0 if $j$ is odd. Hence,

$$
\int_{\mathbb{R}} \frac{\left(1 + \frac{(y+a)^2}{d}\right)^{\frac{d+1}{2}}}{\left(1 + \frac{y^2}{d}\right)^{d+1}} \mathrm{d}y = \sum_{l=0}^{\frac{d+1}{2}} \sum_{j=1}^{l} \binom{\frac{d+1}{2}}{l} \binom{2l}{2j} \frac{a^{2l-2j}}{d^l} \int_{\mathbb{R}} \frac{y^{2j}}{\left(1 + \frac{y^2}{d}\right)^{d+1}} \mathrm{d}y
$$

Then, we compute the integrals. By change of variable $u = y/d$, we have

$$
\int_{\mathbb{R}} \frac{y^{2j}}{\left(1 + \frac{y^2}{d}\right)^{d+1}} \mathrm{d}y = d^{j+1/2} \int_{\mathbb{R}} \frac{u^{2j}}{(1 + u^2)^{d+1}} \mathrm{d}u \le 2 d^{j+1/2}
$$

and for $l = j$,

$$
\sum_{l=0}^{\frac{d+1}{2}} \binom{\frac{d+1}{2}}{l} \frac{1}{d^l} \int_{\mathbb{R}} \frac{y^{2l}}{\left(1 + \frac{y^2}{d}\right)^{d+1}} \mathrm{d}y = \int_{\mathbb{R}} \frac{(1 + y^2/d)^{\frac{d+1}{2}}}{\left(1 + \frac{y^2}{d}\right)^{d+1}} \mathrm{d}y
$$

Hence,

$$
\int_{\mathbb{R}} \frac{\left(1 + \frac{(y+a)^2}{d}\right)^{\frac{d+1}{2}}}{\left(1 + \frac{y^2}{d}\right)^{d+1}} \mathrm{d}y \le 2 \sum_{l=1}^{\frac{d+1}{2}} \sum_{j=0}^{l-1} \binom{\frac{d+1}{2}}{l} \binom{2l}{2j} \frac{a^{2l-2j}}{d^l} d^{j+1/2} + \int_{\mathbb{R}} \frac{(1 + y^2/d)^{\frac{d+1}{2}}}{\left(1 + \frac{y^2}{d}\right)^{d+1}} \mathrm{d}y
$$

$$
= 2 \sum_{l=1}^{\frac{d+1}{2}} a^{2l} \sum_{j=0}^{l-1} \binom{\frac{d+1}{2}}{l} \binom{2l}{2j} \frac{a^{-2j}}{d^l} d^{j+1/2} + \int_{\mathbb{R}} \frac{(1 + y^2/d)^{\frac{d+1}{2}}}{\left(1 + \frac{y^2}{d}\right)^{d+1}} \mathrm{d}y
$$

$$
\le 2 \sum_{l=1}^{\frac{d+1}{2}} a^{2l} \sum_{j=0}^{l-1} \binom{\frac{d+1}{2}}{l} \binom{2l}{2j} \frac{a^{-2j}}{d^l} d^{j+1/2} + \int_{\mathbb{R}} \frac{(1 + y^2/d)^{\frac{d+1}{2}}}{\left(1 + \frac{y^2}{d}\right)^{d+1}} \mathrm{d}y \tag{45}
$$

$$
\tag{46}
$$

And,

$$
\sum_{l=1}^{\frac{d+1}{2}} a^{2l} \sum_{j=0}^{l-1} \binom{\frac{d+1}{2}}{l} \binom{2l}{2j} \frac{a^{-2j}}{d^l} d^{j+1/2} = \sqrt{d} \sum_{l=1}^{\frac{d+1}{2}} \sum_{j=0}^{l-1} \binom{\frac{d+1}{2}}{l} \binom{2l}{2j} a^{2(l-j)} d^{j-l}
$$

$$
\le \sqrt{d} \sum_{l=1}^{\frac{d+1}{2}} \sum_{j=0}^{l-1} \binom{\frac{d+1}{2}}{l} \binom{2(l-1)}{2j} l^2 \left(\frac{a^2}{d}\right)^{l-j}.
$$

Using that $\binom{2l}{2j} = \binom{2(l-1)}{2j} \frac{2l(2l-1)}{(2l-2j)(2l-2j-1)} \le \binom{2(l-1)}{2j} l^2$.

Then, completing the binomial sum so that

$$\sum_{j=0}^{l-1} \binom{2(l-1)}{2j} \left(\frac{a^2}{d}\right)^{-j} \le \sum_{j=0}^{2(l-1)} \binom{2(l-1)}{2j} \left(\frac{a^2}{d}\right)^{-j} = \left(1 + \frac{\sqrt{d}}{a}\right)^{2(l-1)},$$

we have,

$$\sum_{l=1}^{\frac{d+1}{2}} a^{2l} \sum_{j=0}^{l-1} \binom{\frac{d+1}{2}}{l} \binom{2l}{2j} \frac{a^{-2j}}{d^l} d^{j+1/2} \le \frac{1}{2}(d+1)\sqrt{d} \sum_{l=1}^{\frac{d+1}{2}} \binom{\frac{d+1}{2}}{l} \left(\frac{a^2}{d}\right)^l l \left(1 + \frac{\sqrt{d}}{a}\right)^{2(l-1)}$$

$$= \frac{a^2}{2d}(d+1)\sqrt{d} \sum_{l=1}^{\frac{d+1}{2}} \binom{\frac{d+1}{2}}{l} l \left(\frac{a}{\sqrt{d}} + 1\right)^{2(l-1)}$$

$$= \frac{a^2}{2d}(d+1)\sqrt{d} \sum_{l=0}^{\frac{d-1}{2}} \binom{\frac{d-1}{2}}{l} \frac{(d+1)(l+1)}{2(l+1)} \left(\frac{a}{\sqrt{d}} + 1\right)^{2l}$$

$$\le \frac{a^2}{4\sqrt{d}}(d+1)^2 \left(2 + \frac{a}{\sqrt{d}}\right)^{d-1}$$

Then, inject this in Equation (45) to get

$$\int_{\mathbb{R}} \frac{\left(1 + \frac{(y+a)^2}{d}\right)^{\frac{d+1}{2}}}{\left(1 + \frac{y^2}{d}\right)^{d+1}} \mathrm{d}y \le \frac{a^2}{2\sqrt{d}}(d+1)^2 \left(2 + \frac{a}{\sqrt{d}}\right)^{d-1} + \int_{\mathbb{R}} \frac{(1 + y^2/d)^{\frac{d+1}{2}}}{\left(1 + \frac{y^2}{d}\right)^{d+1}} \mathrm{d}y.$$

# C   Additional experimental results

## C.1   Sensitivity to $\beta$ and $\varepsilon$

In this section, we illustrate the impact of the choice of $\beta$ and $\varepsilon$ on the estimation.

**Choice of $\beta$ (Figure 5(b)):**   The choice of $\beta$ is a trade-off between the bias (distance $|\mathrm{Hub}_\beta(P) - \mathbb{E}[X]|$ which decreases as $\beta$ go to infinity) and robustness (when $\beta$ goes to 0, $\mathrm{Hub}_\beta(P)$ goes to the median). To illustrate this trade-off we use the Weibull distribution for which can be very asymmetric. We use a 3-armed bandit problem with shape parameters $(2, 2, 0.75)$ and scale parameters $(0.5, 0.7, 0.8)$ which implies that the means are approximately $(0.44, 0.62, 0.95)$. These distributions are very asymmetric, hence the bias $|\mathrm{Hub}_\beta(P) - \mathbb{E}[X]|$ is high and in fact even though arm 3 has the optimal mean, arm 2 will have the optimal median, the medians are given by $(0.41, 0.58, 0.49)$. In this experiment we don't use any corruption as we don't want to complicate the interpretation. As expected by the theory, we get that $\beta_i$ should not be too small or too large but it should be around $4\sigma_i$.

**Choice of $\varepsilon$ (Figure 5(a)):**   To illustrate the dependency in $\varepsilon$, we also use the Weibull distribution to show the dependency in $\varepsilon$ with the same parameters as in the previous Weibull example, except that we choose $\beta_i = 5\sigma_i$ which is around the optimum found in the previous experiment and we corrupt with $2\%$ of outliers (this is the true $\varepsilon$ while we will make the $\varepsilon$ used in the definition of the algorithm vary). The outliers are constructed as in Section 7. The effect of the parameter $\varepsilon$ is difficult to assess because $\varepsilon$ has an impact on the length of force exploration that we impose at the beginning of our algorithm (the $s_{lim}$).

Figure 4: Cumulative regret plots for different values of the parameters $\varepsilon$ and $\beta$ on a Weibull dataset.

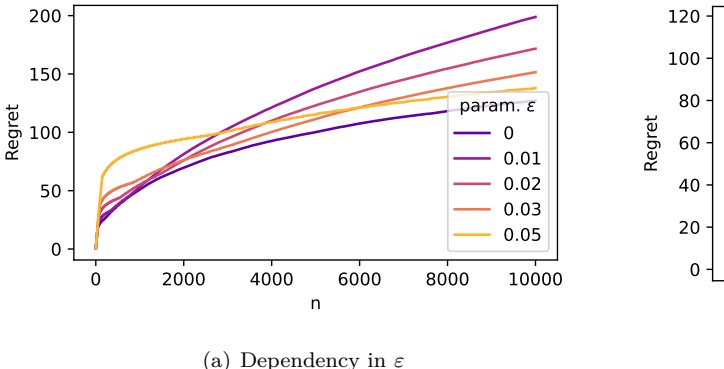

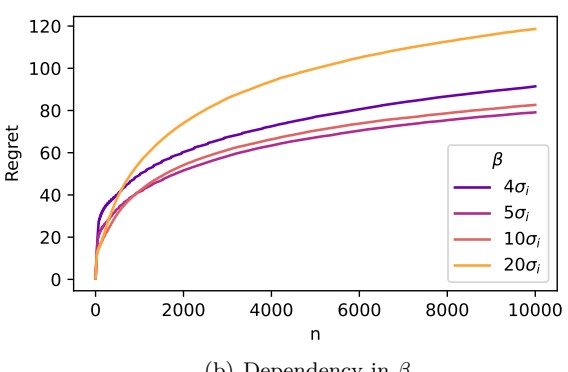

(a) Dependency in $\varepsilon$                       (b) Dependency in $\beta$

## C.2   Corrupted bandits with adversarial algorithms

To illustrate the performances of classical algorithms on corrupted bandits problems, we redo the experiments from Section 7 with algorithms from the adversarial literature (Exp3 and FTRL with log-barrier). We also include Thompson sampling in the case of Bernoulli inlier distributions. The results are rendered in Figure 5. These results show that adversarial algorithms like EXP3 and FTRL, and also Thompson Sampling are very inefficient when the corruption is important as in the case of the Pareto experiments with $\varepsilon = 0.03$ and $\varepsilon = 0.05$.

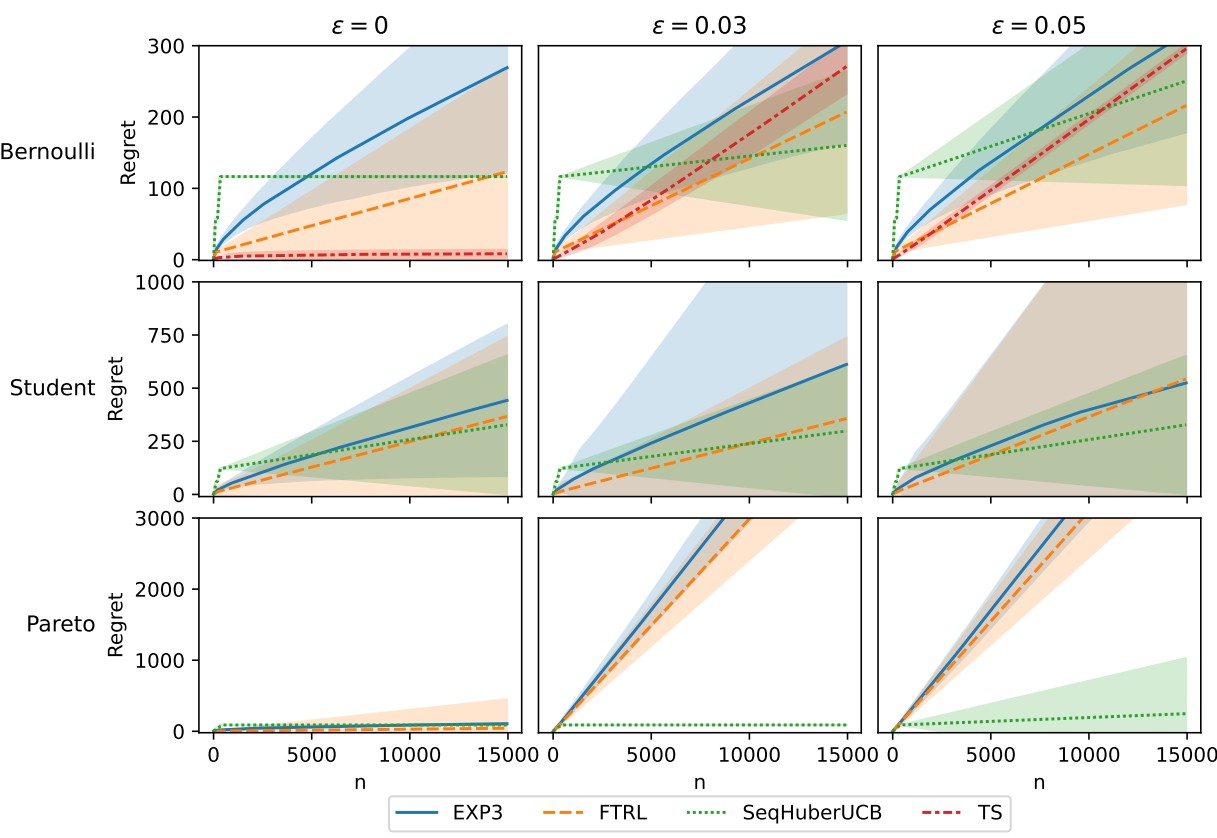

Figure 5: Cumulative regret plot of the algorithms on a corrupted Bernoulli (above), Student's (middle) and Pareto (below) reward distributions with various corruption levels $\varepsilon$. Lower corrupted regret indicates better performance for an algorithm.

