# OpenReview forum: "Bandits Corrupted by Nature: Lower Bounds on Regret and Robust Optimistic Algorithms"
_TMLR — Accepted by TMLR_

### Review · Reviewer_CD4D · 2022-11-21

**Summary Of Contributions:**

This paper studies multi-armed bandits with stochastic rewards and stochastic corruption with an explicit focus on heavy-tailed distribution.
They leverage the concentration of Huber's robust estimator to produce a UCB style algorithm and prove its upper bound.
The i.i.d. nature of the corruption makes this an instance of bandits with distribution shift, where the observations are sampled from a different distribution than the actual rewards/regret.

**Audience:**

Yes

**Broader Impact Concerns:**

No concern.

**Claims And Evidence:**

Yes

**Requested Changes:**

Necessary changes
* Rework the proof of Theorem 1, which relies right now on a Lemma that only holds for centered Bernoulli random variables.
* The authors should run a spellcheck since there are avoidable spelling mistakes such as 'tge' -> 'the' in the paper and also several grammar mistakes.

Recommended changes
* replace the Bernoulli in the corrupted lower bound example with an actual heavy tailed example.
* extend the upper bound to unknown epsilon
* include logbarrier based ftrl in the experimental section



**Strengths And Weaknesses:**

The estimator and its concentration are interesting, as well as the general problem setup.

However, I feel that the current work is not giving a clear picture of the actual difficulty of the problem. The main weakness is in the lower bound section. There is a lower bound for the uncorrupted heavy tail setting, and a lower bound for the corrupted Bernoulli case. The latter one is in my opinion completely redundant and not of interest. Looking at the problem from the distribution shift perspective, the instance dependent worst-case corruption bound is obviously when the optimal arm is shifted towards less rewards and all other arms towards highest rewards. If the identity of the optimal arm changes, then no-regret is impossible. Otherwise classical results yield directly asymptotically tight lower bounds.
In this work, they are trying to reframe the bounds in terms of gap and variance. Theorem 1 for Corrupted Bernoulli is stated too strongly because they only prove the lower bound on the KL divergence for pairs of Bernoulli random variables that are symmetric around 1/2.

For the upper bound, a clear weakness is a lack of discussion about how to potentially overcome a lack of knowing epsilon, since it seems unrealistic to know the mixing parameter in practice.

Finally the empirical section is not very convincing. The Bernoulli case is not of interest because the best you can do is just ignore potential corruption and run e.g. Thompson Sampling (since corruption is undetectable and we do not need to hedge for it). For the adversarial baseline, it would be better to use an algorithm that can handle heavier tails such as log-barrier based methods.

Edit after reviewing TMLR policy:
Besides the issue of stating Theorem 1 too strongly since the corresponding Lemma only holds for special pairs of Bernoulli random variables, there are no other clear technical flaws in this paper.
I would still suggest the changes proposed in my original review to increase the significance of this paper, but I understand that this is not a criteria for this Journal.

---

> ### Author Response · Authors · 2023-02-16
> **Response to reviewer**
>
>
> We thank the reviewer for his/her critical reading of our work. Hereby, we aim to address the concerns.
>
> - *Bernoulli lower bound:* The reviewer stated: "There is a lower bound for the uncorrupted heavy tail setting, and a lower bound for the corrupted Bernoulli case. The latter one is in my opinion completely redundant and not of interest". We do not understand why the corrupted Bernoulli lower bound is redundant as it is not a consequence of the lower bound for Student distributions. Further, not such bound for corrupted bandits have been proposed so far. The proposed lower bounds are for specific distribution.   Our goal is to get an idea about the hardness regimes induced by corruption and also the achievable order of regret w.r.t. $T$. The proposed lower bounds serve this purpose. We disagree that this is a weakness, we believe it is important to provide such intuition while deriving a general interpretable lower bound in a heavy-tail and corrupted setting remain an open problem. "Theorem 1 for Corrupted Bernoulli is stated too strongly because they only prove the lower bound on the KL divergence for pairs of Bernoulli random variables that are symmetric around 1/2." We do not understand this statement as there is only one Bernoulli distribution which is symmetric around 1/2 (i.e. a Bernoulli with mean 1/2) and this is not what we use.
>
> - *Deriving lower bound from known results:*
> We are not studying the distribution shifts as they are studied in non-stationary bandits. In our setting, the underlying reward distributions remain invariant over time. The rewards generated from them are corrupted using unbounded corruption distributions before being observed by the algorithm. Using robust estimators, still we aim to identify the optimal arm and the sub-optimal ones in the sense of "true" expected reward while observing only the "corrupted" rewards.
>
> - *Knowledge of $\varepsilon$:* We thank the reviewer for raising this point. We refer to our response to a similar question raised by Reviewer YBeD for detailed discussion.
>
> - *Experiments:* Using Thompson Sampling does not seem to be optimal, even in the case of Bernoulli. Indeed, the corruption may cause to exchange the order of the means perceived by a non-robust estimator (i.e. the sample mean) while not changing the order for robust estimator (i.e. Huber estimator). Hence, the regret can go linear for a non-robust algorithm, and logarithmic when using Huber estimator. Experimental results validate this theoretical intuition. We will include experimental results of Thompson-Sampling and log-barrier based FTRL algorithms in the final version for further validation.
>
> We will also proofread the draft to edit all the typos.

---

> > ### Comment · Reviewer_CD4D · 2023-03-01
> > **Clarification lower bound**
> >
> > "We do not understand this statement as there is only one Bernoulli distribution which is symmetric around 1/2 (i.e. a Bernoulli with mean 1/2) and this is not what we use."
> >
> > What I mean is that Theorem 1 for the Bernoulli case uses Lemma 4, which only applies to pairs of Bernoulli random variables that satisfy $P(X_0=1)=P(X_1=0), P(X_0=0)=P(X_1=1) $. Hence "centred around 1/2", since $E[0.5(X_0+X_1)] = 0.5$.
> >
> > That means the statement in the theorem (for any arm $i$), is actually only proven for any arm $i$ such that $E[X_i] = 1-E[X^*]$, which is not what this theorem statement suggests.

---

> > > ### Author Response · Authors · 2023-03-01
> > > **Response**
> > >
> > > Thanks for the clarification.
> > >
> > > Through our lower bound of the Bernoulli case, we are trying to understand how hard it can be to distinguish two Bernoulli distributions under the worst-case corruption. Thus, Lemma 4 is an existence result, where we aim to upper bound the KL divergence between any two Bernoulli distributions. To be specific, since for any $P_1,P_2 \in  \mathrm{Bernoulli}$ and corresponding corruption distributions $H_1$ and $H_2$, we have
> > > $$ D_{\mathrm{KL}}((1-\varepsilon)P_1+\varepsilon H_1,(1-\varepsilon)P_2+\varepsilon H_2) \le \sup_{P_1,P_2 \in  \mathrm{Bernoulli}, H_1,H_2 \in \mathcal{P} }D_{\mathrm{KL}}((1-\varepsilon)P_1+\varepsilon H_1,(1-\varepsilon)P_2+\varepsilon H_2)  $$
> > > In Lemma 4, we prove an upper bound on the RHS by showing that there exists $P_1^{\sup},P_2^{\sup}$ and $H_1^{\sup},H_2^{\sup}$ for which the Equations (3) and (4) hold true.
> > > Thus, though $P_1^{\sup}$ and $P_2^{\sup}$ may abide by certain constraints, the corresponding upper bounds still indicate the hardness of the problem.
> > >
> > > We will add this clarification to the final version of the paper.

---

### Review · Reviewer_YBeD · 2022-12-27

**Summary Of Contributions:**

This paper considers a kind of stochastic multi-armed bandit problem with heavy tailed reward distributions (with bounded moments) and unbounded corruptions. In the problem, it is assumed that the reward comes from an unknown fixed corruption distribution distribution with an unknown fixed probability. For this problem, problem-dependent lower bounds are shown. Further, this paper proposes a new UCB-type algorithm that achieves a nearly optimal regret bound. A variant with improved computational efficiency is also proposed.

**Audience:**

Yes

**Claims And Evidence:**

Yes

**Requested Changes:**

Regarding Huber's estimator, I think it is difficult to understand, in the current notation $Hub(\cdot)$, that its value also depends on $\beta$.
In particular, the fact that $\beta_i$ is used in the definition of $Hub_{i,s}$ and $B_i$ in HuberUCB is difficult to read from the definition formula.
I belive that there is room for improvement in notation or explanation for the sake of clarity.


Minor comment:

There were several notations that were not consistent, e.g.,
- subGaussian vs sub-Gaussian,
- $P_{P_0}$ vs $\mathbb{P}_{P_0}$ (between Eq. (2) and Eq. (3))
- $D_{KL}$-divergence and KL-divergence

The following may be typos:
- i.i.d <- i.i.d.
- Algorithm 2: require $\beta$ <- $\\{ \beta_i \\}$
- Theorem 3: $\beta \leq 4 \sigma_i$ <- $\beta_i \leq 4 \sigma_i$
- p.14: tge true <- the true

Some of the references cite preprints even though have already been published (e.g., [Lattimore and Szepesvári, 2018] <- [Lattimore and Szepesvári, 2020]), so it would be better to replace them with the published version.

**Strengths And Weaknesses:**

Strength:
- The motivation for considering the proposed model and its relevance to existing research is carefully explained.
- New techniques have been proposed, such as a concentration inequality for the empirical Huber's estimator (Theorem 2) and a faster sequential approximation algorithm for this computation (Section 6), which have potential applications to problems not limited bandits.
- Lower and upper bounds for regret nicely depict how parameters (such as suboptimality gap $\Delta$, variance $\sigma^2$ and corruption level $\epsilon$) affect the difficulty of the problem, which should be of interest of some individuals in the comunity of the learning theory.
- Claims appear to be supported by convincing and clear evidence. As far as I could see, there were no errors in the analysis or unclear definitions.

Weakness:
- In the design of the algorithm, it is assumed that the variance $\sigma_i^2$ of the reward and the rate $\epsilon$ of corruption are known in advance. This assumption poses a major limitation in applications as these parameters are often not known a priori in practice.

In view of the above strength, and in light of the TMLR evaluation criteria, I support the acceptance of this paper.

---

> ### Author Response · Authors · 2023-02-16
> **Response to reviewer**
>
> We thank the reviewer for appreciating the new techniques and results proposed in the paper while aptly pointing out the limitations of the present assumptions. Here, we address your concerns.
>
> - *Unknown variance:* A way to handle unknown variance is to use Empirical Bernstein techniques (ref. ``Empirical Bernstein Bounds and Sample Variance Penalization", Maurer, A., and Pontil, M., COLT 2009). In our setting, it would be necessary to have a robust empirical Bernstein estimate which would in particular entail the need to have a robust estimator of the variance. This approach is technical and would complicate the proof and we feel that this may muddle the message of the paper. But we agree that designing such an algorithm will be an interesting and realistic research work.
>
> - *Unknown corruption proportion:* Estimating the proportion of corruption has been done, e.g. using proportion of point larger than 1.5 times the inter-quartile range or using more complex algorithms like Isolation Forest algorithm and similar anomaly detection algorithms. To the best of our knowledge, the techniques remain in the realm of heuristics and estimating $\varepsilon$ is in general hard in a non-parametric setting, at least it is not something classical in the Robust Statistics literature. Moreover, on a maybe more conceptual side, the problem of estimating $\varepsilon$ is not really a well-posed problem in this setting, as we do not make any assumptions on the outlier distribution. For example, in all the works on estimation of $\varepsilon$ that we are aware of, the outliers are usually defined as point that are outside of the bulk of the data in some sense which is a restriction compared to our setting. Moreover even though there are heuristics, the problem of finding what constitute "the bulk of the data" is closely linked to problems such as finding a ``Robust minimum volume ellipsoid". This is NP-hard in general (c.f. "Finding Minimum Volume Circumscribing Ellipsoids Using Generalized Copositive Programming" Mittal and Hanasusanto (2018)). We add a discussion explicating the hardness of estimating $\varepsilon$ in the Section 5.4. of the revised draft.
>
> - *Requested Changes:* We revise the notations to make the dependence of Huber estimator on $\beta$ more explicit. We change $Hub$ to $Hub_{\beta}$ but keep $Hub_{i,s}$ for simplicity of notation.
>
> - *Minor comment:* We thank the reviewer for his careful reading, we will incorporate all the suggested changes in the final version of the article.

---

### Review · Reviewer_52bW · 2023-02-04

**Summary Of Contributions:**

This paper examines the bandit problem in a stochastic corruption setting, also known as the "corrupted by nature" setting. In this setting, the bandit feedback has a low probability of being drawn from a different, potentially heavy-tailed distribution. The paper first establishes a problem-specific lower bound based on the corrupted gap of the rewards. Then, it introduces the HubUCB algorithm, a UCB-type solution that uses a Huber estimator to address heavy-tailed corruption. The paper demonstrates that the algorithm attains a near-optimal regret upper bound. Finally, the paper enhances the computational efficiency of the Huber estimator by presenting a sequential version of the estimator, which exhibits linear computational complexity.

**Audience:**

Yes

**Claims And Evidence:**

Yes

**Requested Changes:**

- Perhaps modify the motivating example in intro?
- Make core concepts well-defined, such as the heavy-tailedness.
- Def. 1: what is the range of $n$? Should it be any $n>0$?
- Last paragraph of page 6, what is $\mathcal{P}_{[2]}$?
- Was this paper https://proceedings.neurips.cc/paper/2021/file/843a4d7fb5b1641b0bb8e3c2b2e75231-Paper.pdf somehow related to your heavy-tail setting?

**Strengths And Weaknesses:**

Strength:
- A systematical study of the "corrupted by nature" setting, including proposing a new notion of "corrupted regret" is introduced.
- Problem-dependent lower-bounds and upper-bounds for the problem were proposed.
- An algorithm with improved computation complexity is provided.
- A numerical study demonstrates the effectiveness of the proposed algorithm.

Weakness:
- The major weakness appears in the presentation. In the intro, the "Treatments of Varroa Mites" does not seem to be a good example. First, it is hard to argue that the distribution is heavy-tailed. Secondly, why corruption can be unbounded? Moreover, this kind of task usually only depends on the human experience. It is hard to believe that the method proposed in this paper would help solve this problem quantitatively.
- No bounds on the problem-independent setting.
- The upper bounds were claimed to be nearly tight, but unfortunately, the gap parameters are different in the lower bound.
- Problem-dependent lower bounds do not work for all cases, but only for special cases (student-T and Bernoulli).
- Heavytailedness is not well-defined. What is the condition that makes the distribution heavy-tailed? In the lower bound, only student distribution is used for heavy-tailedness. In the upper bound,  the first time the heavy-tailedness is discussed is in the lemma about the bias of Huber estimator. If heavy-tailedness is a major theme of this paper, it should probably be clearly defined in the preliminary section of the paper and also reflect this, perhaps, in the title.

---

> ### Author Response · Authors · 2023-02-16
> **Response to reviewer**
>
> Thanks for your constructive comments and appreciating the contributions. Hereby, we address your concerns.
>
> - *Motivating example:* The motivating example is meant as an illustration of how corruption could appear in a decision problem, the proposed algorithm is meant to be a methological contribution and  not to be directly applied to the problem. This would indeed involve several other scientific challenges.
> Regarding the assumption, the number of Varroa Mites can be rather large and even though we agree that it is not unbounded, it is our opinion that it is better to model this with unbounded distribution because the behaviour of a ``very variable" distribution as it is called by the biologists in the articles on the subject, is better modelled using unbounded/heavy-tailed distributions. See the papers on robust statistics by Tukey "A survey of sampling from contaminated distributions" (1960) and the first chapter of the book "Robust Statistics" (2004) by Huber.
> Thus, following the long-studied robust estimation literature, we propose a theoretical formulation of the real-world problem in terms of bandits corrupted by nature that captures the hardness evoked by imperfect observations.
>
> - *No bound in the problem independent setting:* Thanks for the suggestion. We felt that a data dependent approach was better to give intuition on the constants really involved in the problem (i.e. this shifted gap parameter), which would not be visible in a worst-case approach. Still, deriving a problem-independent regret bound will be an interesting future work. We will add it to the Future Work section.
>
> - *Tightness of the results:* The results are only rate-optimal (i.e. the bound are logarithmic in $n$) but indeed they are not tight. We refer to the discussion on page 13 for more information on this. Additionally, we edit the text to clarify that by tightness, we mean rate-optimality.
>
> - *Limitations of the proposed lower bounds:* The proposed lower bounds are indeed restricted to specific distributions. The general problem of finding an explicit lower bound in a corrupted setting is not trivial as there is a need to optimize on the whole corruption neighborhood in the bound obtained in Lemma 2. As we have mentioned in the Future Work, deriving such bounds is still an open problem in a non-parametric setting. We chose to present the two specialized lower bounds in order to provide the intuitions regarding low vs high distinguishibility as well as of the shifted gap from Section 4.
>
> - *Heavy tail distributions:* Thanks for pointing out. A heavy-tailed distribution is a standard notion, defined as a distribution that have a finite number of finite moments (as opposed to sub-Gaussian or sub-Gamma distributions). This is a popular way of calling the distributions belonging to one of the sets $\mathcal{P}_{[q]}$. We will clarify this in the introduction of the revised article.
>
> - *Def. 1: what is the range of $n$ ? Should it be any $n>0$ ?:*
> Definition $1$ is an asymptotic concept, which extends the classical concept of uniformly good policies in bandits. We ask that the regret of the algorithm is a $o(n^{\alpha})$, when $n$ goes to infinity, as such $n$ is supposed to be in a neighborhood of infinity (Big Omicron and big Omega and big Theta, Donald E. Knuth. ACM SIGACT News, 1976). We will clarify this.
>
> - *Last paragraph of page 6, what is $\mathcal{P}_{[2]}$?:*
> $\mathcal{P}_{[2]}$ is the set of distributions with a finite second moment. We have defined in the Notations paragraph in Section 3 (at the bottom of page $4$).
>
> - *Was this paper https://proceedings.neurips.cc/paper/2021/file/843a4d7fb5b1641b0bb8e3c2b2e75231-Paper.pdf somehow related to your heavy-tail setting?:*
> Yes the mentioned paper is related. The authors consider finite $1+\varepsilon$ moment for $\varepsilon \in (0,1)$, in our article we consider $2$ finite moments. We will add the citation in revised draft.

---

### Author Response · Authors · 2023-02-16
**General comment**

We thank the reviewers for their constructive feedback, and
editing and proofreading suggestions, which we will take into account for the final version. In the following, we respond to the technical comments of the reviewers individually.

---

### Decision · Action_Editors · 2023-05-04

**Recommendation:** Accept with minor revision

**Comment:**

There are a number of presentation issues which need to be addressed before the final version. A (partial) list is provided below, please make sure to correct all of them and go through the paper again to make similar improvements where necessary.

Technical issues: The writing is not always clear, some definitions are missing and the use of notation is not always clear. Some examples are given below:
- In the problem formulation $\nu$ and $P_i$ seem to be connected, defining the environment. Then in Lemma 2, they are selected independently, and it is not specified which one of them defines now the environment. First $\nu$ is fixed, then the statement is for "any $P_i$", followed by the $\nu$ being a free variable in the definition of $\mathcal{K}$. Neither $P^*$ nor $D_{KL}$ seem to be defined.
In the discussion following the lemma, $P_i$ (and $P^*$) are considered as "arms", while they are reward distributions associated with these arms.
- Student distributions  have zero mean, and so is the set $\mathcal{T}_d$, but Lemma 3 and the theorems use shifted student distributions.
- What is $\delta_c$ in the corrupted Bernoulli definition? By the way, I think it would be clearer to say that you consider the family of corrupted Bernoulli distributions as defined above Theorem 1, and then say you consider two elements from this (does this mean $c=1$ in $Q_0$?). Introducing the corrupted Bernoulli with pairs of distributions is a bit confusing.
- Lemma 4 is again a bit confusing: Are $Q_0$ and $Q_1$ the same as defined above the lemma (i.e., with an atom added to $Q_1$ at 0)? If yes, it would be simpler to define them in the lemma. Is there a special meaning to "shifted suboptimality gap"? I assume not, in which case it might be clearer to just say suboptimality gap. In any case, the lemma is very strangely phrased as $\sigma$ and $\Delta$ fully determine $P_0$ and $P_1$ (thus the "there exists" is not really appropriate), and it is also not clear from the statement that the $Q_i$ are Bernoulli. Please define everything properly (e.g., given $\sigma$ and $\Delta$, define $P_0$ and $P_1$, etc.). You may want to say that $\Delta>0$.
- Theorem 1: It is never explicitly mentioned that in the Student/Bernoulli distributions cases all reward distributions are Student/Bernoulli (in facg it might be best to introduce these settings properly when you give the corresponding KL bounds). In the Bernoulli case it might be worth mentioning the exact form of $P_i$. The sign of $\Delta_i$ is flipped.
- Theorem 3/Corollary 1: You should mention explicitly which algorithm is run.
- p. 5: "Assuming a non-adversarial behavior of the Nature seem" -> seems
- Citations: Please make sure your use of citations is correct (see the "Citations" section in https://www.jmlr.org/format/format.html and https://www.jmlr.org/format/formatting-errors.html).

The following suggestions should also be considered for the final version:
- The name "corrupted by nature" is a bit mysterious, in the sense the role of "nature" is not clear from the wording. Calling it something like "bandits with (stochastic) corruption" seems to be more descriptive. (Also, a "corrupted bandit algorithm" is a bandit algorithm which is corrupted, you may want to use something like "an algorithm for the corrupted-bandit problem" or a "corrupted-bandit algorithm.")
- The reviewers were not very happy with the motivating example, so coming up with some others (where actually the number of decisions is in the ballpark of what is used in the experiments) would be of interest.
- To me the name "corrupted regret" also seems a bit misleading as in some sense the quantity is more like the non-corrupted regret (i.e., the regret the algorithm would suffer on the non-corrupted part of the data), but I understand that the motivation behind this naming is that the observations are corrupted.



**Audience:**

The reviewers found that the claims in the paper are correct and the results are novel enough to gain interest of a small part of the community. None of the reviewers were especially excited, but some members of the community may find the setting or the technical results interesting.

**Claims And Evidence:**

This paper examines the multi-armed bandit problem in a stochastic corruption setting. In this setting, with low probability the bandit feedback is drawn from a different, potentially heavy-tailed distribution (corruption). The problem is analyzed by presenting a problem dependent lower bound on the regret of any algorithm, as well as an UCB-algorithm achieving near optimal regret. A computationally more efficient variant of the algorithm is also derived, and simple experiments complement the theoretical findings.

---

> ### Author Response · Authors · 2023-07-13
> **Answer to the comments of the Action Editor(s)**
>
> Dear Action Editor(s) and Editor(s),
>
> We have updated the camera-ready version in answer to your comments.
> About the citations, we used the tmlr bibliography style advised in the template provided by TMLR. We also corrected some misuses of citep vs cite, we hope this will answer your concerns.
>
> Thanks, Authors of paper 530